# Non-parametric classification via expand-and-sparsify representation

**Kaushik Sinha**
School of Computing
Wichita State University
Wichita, KS 67260
kaushik.sinha@wichita.edu

## Abstract

In *expand-and-sparsify* (EaS) representation, a data point in $\mathcal{S}^{d-1}$ is first randomly mapped to higher dimension $\mathbb{R}^m$, where $m > d$, followed by a sparsification operation where the informative $k \ll m$ of the $m$ coordinates are set to one and the rest are set to zero. We propose two algorithms for non-parametric classification using such EaS representation. For our first algorithm, we use *winners-take-all* operation for the sparsification step and show that the proposed classifier admits the form of a locally weighted average classifier and establish its consistency via Stone's Theorem. Further, assuming that the conditional probability function $P(y = 1|x) = \eta(x)$ is Hölder continuous and for optimal choice of $m$, we show that the convergence rate of this classifier is minimax-optimal. For our second algorithm, we use *empirical $k$-thresholding* operation for the sparsification step, and under the assumption that data lie on a low dimensional manifold of dimension $d_0 \ll d$, we show that the convergence rate of this classifier depends only on $d_0$ and is again minimax-optimal. Empirical evaluations performed on real-world datasets corroborate our theoretical results.

## 1 Introduction

Given a training set $x_1, \ldots, x_n$ of size $n$ and a test point $x$ sampled *i.i.d.* from a distribution $\mu$ over some sample space $\mathcal{X} \subset \mathbb{R}^d$, the basic idea of non-parametric estimation (e.g., density estimation, regression or classification) is to construct a function $x \mapsto f_n(x) = f_n(x, x_1, \ldots, x_n)$ that approximates the true function $f$ (which could be a density/regression/classification function as appropriate) as $n \to \infty$ Tsybakov [2008]. For any $x \in \mathcal{X}$, $f_n(x)$ typically depends on a small subset of the training set lying within a small neighborhood of (thus close to) $x$. For example, in case of $k$-nearest neighbor, $f_n(x)$ depends on the $k$ points from the training set that are closest to $x$; in case of random forest, for each tree constituting the forest, $f_n(x)$ depends on the points from the training set that are routed to the same leaf node to which $x$ is also routed to; in case of kernel methods, $f_n(x)$ depends on the points from training set defined by an appropriate kernel. In this paper, we propose a new non-parametric classification method where, for any $x \in \mathcal{S}^{d-1}$, the appropriate neighborhoods are obtained using expand-and sparsify (EaS) representation of $x$.

On a high level, expand-and-sparsify representation is essentially a transformation from a low-dimensional dense representation of sensory inputs to a much higher-dimensional, sparse representation. Such representation has been found, for instance, in the olfactory system of the fly (Wilson [2013]) and mouse (Stettler and Axel [2009]), the visual system of the cat (Olshausen and Field [2004]), and the electrosensory system of the electric fish (Chacron et al. [2011]). For example, in the olfactory system of Drosophila (Turner et al. [2008], Masse et al. [2009], Wilson [2013], Caron et al. [2013]), the primary sense receptors of the fly are the roughly 2,500 odor receptor neurons

38th Conference on Neural Information Processing Systems (NeurIPS 2024).

(also known as, ORNs), which can be clustered into 50 types, based on their odor responses, leading to a dense, 50-dimensional sensory input vector. In fact, all ORNs of a given type converge on a corresponding glomerulus in the antennal lobe; there are 50 of these in a topographically fixed configuration. This information is then relayed via projection neurons to a collection of roughly 2000 Kenyon cells (KCs) in the mushroom body, with each KC receiving signal from roughly 5-10 glomeruli (Dasgupta and Tosh [2020]. The pattern of connectivity between the glomeruli and Kenyon cells appears random (Chacron et al. [2011]). The output of the KCs is integrated by a single anterior paired lateral (APL) neuron which then provides negative feedback causing all but the 5% highest-firing KCs to be suppressed (Lin et al. [2014]). The result is a random sparse high-dimensional representation of the sensory input, that is the basis for subsequent learning. The primary motivation of this paper is to study the benefit of this type of naturally occurring representation in the context of supervised classification.

In our setting, the EaS representation is a mapping from the $d$-dimensional unit sphere $\mathcal{S}^{d-1}$ to $\{0,1\}^m, m \gg d$, where a data point $x \in \mathcal{S}^{d-1}$ is first randomly mapped to higher dimension $\mathbb{R}^m$ using a random projection matrix $\Theta$, and is followed by a sparsification operation, where the informative $k \ll m$ of the $m$ coordinates of the resulting vector $\Theta x \in \mathbb{R}^m$ are set to one and the rest of the coordinates are set to zero. Rows of $\Theta$ are typically drawn independently from some distribution $\nu$ over $\mathcal{S}^{d-1}$ or $\mathbb{R}^d$. Let $C_j, j = 1, \ldots, m$, be the set of all examples from the input space $\mathcal{X} \subset \mathcal{S}^{d-1}$ whose $j^{th}$ coordinate in respective EaS representations are set to 1. We call $C_j$ the *response region* of $\theta_j$ (the $j^{th}$ row of $\Theta$).

Note that for any $x \in \mathcal{X}$, there are $k$ *activated* response regions corresponding to the $k$ non-zero bits in the EaS representation of $x$. These $k$ activated response regions serves as $k$ neighborhoods of $x$ for our proposed non-parametric classifier, where, for any $x \in \mathcal{X}$ and each activated response region $C_j$, we estimate $\Pr(y = 1|C_j)$ – expected value of $y$ when $x$ is restricted to $C_j$ – and take their average to be the estimate of $\Pr(y = 1|x)$. In a toy example, we visually show the EaS representation and the activated response regions $C_j$ of a data point in Fig. 1. Comparing whether this conditional probability estimate exceeds $1/2$, we predict the class label of $y$. In particular, we present two algorithms using different sparsification schemes and analyze their theoretical properties.

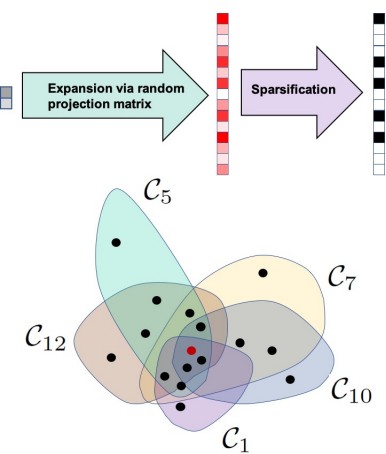

Figure 1: **Top:** A point $x \in \mathbb{R}^2$ (coordinate-wise values are different shades of gray) and its projection $y = \Theta x \in \mathbb{R}^{15}$ (coordinate-wise values are different shades of red). The sparsification step sets the largest 5 values of $y$ to 1 (black squares) and the rest to zero. **Bottom:** Activated response regions $C_j, x \in C_j$, ($x$ is a red dot), are shown using different colors. The points from the training set that intersects with these activated response regions are shown using black dots.

One may find similarity between our proposed algorithm and a random forest classifier – for any $x \in \mathcal{X}$, $k$ activated response regions $C_j, x \in C_j$, may be interpreted as the leaf nodes (containing $x$) of $k$ decision trees in a random forest. However, unlike random forest, we can not simply increase $k$ (number of trees) without changing other hyper-parameters (such as $m$) hoping to achieving better prediction performance. Therefore, even-though the consistency property of random forest is studied under different settings Biau et al. [2008], Scornet et al. [2015], Scornet [2016], Tang et al. [2018], those ideas can not be applied for our theoretical analysis.

We summarize our contributions below:

- We present an interesting connection between non-parametric estimation and EaS representation, and propose two algorithms for non-parametric classification via EaS representation, and present empirical evaluations on benchmark machine learning datasets.

- For our first algorithm (using $k$-WTA sparsification), we establish its universal consistency and prove that it achieves minimax-optimal convergence rate that depends on dimension $d$.

- When data lie on a low dimensional manifold having dimension $d_0 \ll d$, we present a second algorithm (using empirical $k$-thresholding sparsification) and prove that its convergence rate is minimax-optimal and depends only on $d_0$.

The rest of the paper is organized as follows. We discuss related work in 2. We present our first algorithm and its theoretical analysis including consistency and convergence rate in section 3. We present our second algorithm and derive its convergence rate in section 4. We present our empirical results in section 5 and conclude discussing limitations and future work in section 6.

## 2  Related work

Non-parametric estimation is an important branch of statistics with a rich history and classical results. Interested readers may refer to Tsybakov [2008], Györfi et al. [2002], which provide excellent treatment of this subject. Here we briefly review consistency and convergence rates of well-known non-parametric methods such as, partitioning estimates (histograms), $k$-nearest neighbors, kernel methods, and random forests. In the non-parametric literature, it is typical to estimate the regression function $\eta(x) = \mathbb{E}(y|x)$ from data, and use the resulting estimate $\eta_n$ (using a sample of size $n$) to construct plug-in decision function (classification rule). Under mild conditions, such regression estimates and the resulting plug-in classification rules are consistent for histograms, $k$-nearest neighbors, and kernel methods Györfi et al. [2002], Devroye et al. [1996]. Under mild conditions, different variations of random forest methods are also known to be consistent Biau et al. [2008], Scornet et al. [2015], Scornet [2016], Tang et al. [2018]. Under the assumption that the regression function $\eta(x)$ is Lipschitz continuous, the $L_2$ error of the regression estimate of histogram convergences at a rate $O\left(n^{-1/(d+2)}\right)$ (Theorem 4.3 Györfi et al. [2002], the $L_2$ error of the regression estimate of the kernel method convergence at a rate $O\left(n^{-1/(d+2)}\right)$ (Theorem 5.2 Györfi et al. [2002]), and the $L_2$ error of the regression estimate of $k$-nearest neighbors method convergences at a rate $O\left(n^{-2/(d+2)}\right)$ (Theorem 6.2 Györfi et al. [2002]). For random forest, Gao and Zhou [2020] established finite-sample rate $O(n^{-1/(8d+2)})$ on the convergence of pure random forests for classification, which was be improved to be of $O(n^{-1/(3.87d+2)})$ by considering the midpoint splitting mechanism. They introduced another variant of random forests, which follow Breiman's original random forests but with different mechanisms for splitting dimensions and positions, to get a convergence rate $O(n^{-1/(d+2)}(\ln n)^{1/(d+2)})$, which reaches the minimax rate, except for a factor $(\ln n)^{1/(d+2)}$.

For EaS representation, when $\nu$ is the uniform distribution over $\mathcal{S}^{d-1}$ and the sparsification scheme is a $k$-winners-take-all ($k$-WTA) operation that sets the $k$ largest entries of a vector in $\mathbb{R}^m$ to one the rest to zero, Dasgupta and Tosh [2020] proposed a function approximation scheme using such EaS representation that can approximate any Lipschitz continuous function $f : \mathcal{S}^{d-1} \to \mathbb{R}$ in the $L_\infty$ norm, where the error decays exponentially slowly with $d$. Further, they showed when the data lie on a low dimensional submanifold of $\mathcal{S}^{d-1}$ having dimension $d_0 \ll d$, using a different sparsification step, termed as $k$-thresholding, any Lipschiz continuous function defined on this manifold can be approximated in the $L_\infty$ norm, where the error decays exponentially slowly with $d_0$. A different EaS representation, where the projection matrix is a sparse binary matrix and the sparsification step is $k$-WTA, was proposed in Dasgupta et al. [2017] that effectively hash input data points and such hashing has been shown to provide improved performance in accurately solving the similarity search problems compared to the state-of-the-art locality sensitive hashing (LSH) based techniques (Gionis et al. [1999], Andoni and Indyk [2008], Datar et al. [2004]). Such EaS representation has also been used to summarize data in the form of a bloom filter, termed as fly bloom filter (FBF), and has been successfully used in solving the novelty detection problems in Dasgupta et al. [2018] and classification problem in Sinha and Ram [2021], Ram and Sinha [2021, 2022].

While our work is inspired by the results of Dasgupta and Tosh [2020], there are key differences. First, we explicitly expand upon and apply the idea of Dasgupta and Tosh [2020] to the *supervised* binary classification setting, and derive the rate at which the error probability of our proposed classifier converges to that of the Bayes optimal classifier. In comparison, the main motivation of Dasgupta and Tosh [2020] was to prove the existence of a weight vector that results in arbitrarily well function approximation in the *unsupervised learning* setting using EaS representation. Because of this existential nature of their result, the effect of sample size was neither needed nor considered in their result. Second, the $k$-thresholding sparsification scheme proposed in Dasgupta and Tosh

[2020] for function approximation result assumed that the coordinate-wise thresholds were known. We make no such assumption and explicitly describe how to compute such thresholds from data– resulting in realizable algorithm and derive convergence rate of our proposed classifier that takes care of the uncertainly associated with random natures of these thresholds. Third, while Dasgupta and Tosh [2020] only considers LipSchitz continuous functions, we consider that conditional probability function $\eta(x) = \Pr(y = 1|x)$ to be Hölder continuous (a broader function class), and prove that our proposed classifier achieves minimax-optimal convergence rate – whether such optimal convergence rate was achievable in the proposed setting was, unknown prior to our result.

Finally, we note that *random Fourier features* Rahimi and Recht [2007] are generated using a construction similar to `EaS`, however the choice of random directions there are chosen using a kernel function that measures a notion of similarity in the input space. `EaS` representation can also be though of an opposite process of *compressed sensing* Candes et al. [2006], Donoho [2006], where the goal is to recover a sparse vector given random projections to it, while random projection is used to generate a sparse representation in case of `EaS`.

## 3   Algorithm 1

---

**Algorithm 1** Training set $D_n = \{(x_i, y_i)\}_{i=1}^n \subset \mathcal{S}^{d-1} \times \{0, 1\}$, Projection dimensionality $m \in \mathbb{N}$, $k \ll m$ non-zeros in the `EaS` representation, random seed $R$, and inference with test point $x \in \mathcal{S}^{d-1}$.

---

**Train**`EaSClassifier`$(D_n, m, k, R)$
   | Sample $\Theta$ with seed $R$
   | Initialize $w[i], \mathtt{ct}[i] \leftarrow 0, \forall i \in [m]$
   | **for** $(x, y) \in S$ **do**
   |   | $\mathtt{eas} \leftarrow h_1(x)$
   |   | $w[i] \leftarrow w[i] + y, \forall i \in [m] : \mathtt{eas}[i] = 1$
   |   | $\mathtt{ct}[i] \leftarrow \mathtt{ct}[i] + 1, \forall i \in [m] : \mathtt{eas}[i] = 1$
   | **end**
   | $w[i] \leftarrow w[i]/\mathtt{ct}[i], \forall i \in [m]$
   | **return** $\Theta, w$
**end**
**Infer**`EaSClassifier`$(x, \Theta, k, w)$
   | $\mathtt{eas} \leftarrow h_1(x)$
   | **return** $\mathbb{I}[(\mathtt{eas} \cdot w)/k \geq \frac{1}{2}]$
**end**

---

In this section we present our first algorithm and analyze its statistical properties. For the rest of our presentation, we use the following notations. For any positive integer $k \in \mathbb{N}$, we use $[k]$ to denote the set $\{1, \ldots, k\}$. For any vector $v$, we use the notation $v[i]$ to denote its $i^{th}$ coordinate value. We use $\mathcal{S}^{d-1}$ to denote the $d$-dimensional unit sphere and $B(x, r)$ to denote a closed ball of radius $r$ centered at $x$. We use $\mathbb{I}$ to denote indicator variable such that for any boolean variable $A$, $\mathbb{I}[A]$ is 1 if $A$ is `True`, and 0 otherwise. We consider the binary classification setting, where the instance space is $\mathcal{X} \subset \mathcal{S}^{d-1}$ and the output class label is $\mathcal{Y} = \{0, 1\}$. Let $\mu$ denotes the measure on $\mathcal{X}$ and let $\eta : \mathcal{X} \to [0, 1]$, defined as $\eta(x) = \Pr(y = 1|x)$, represents the conditional probability distribution. Then, the joint probability distribution on $\mathcal{X} \times \mathcal{Y}$ is completely defined by $\mu$ and $\eta$. Let $\{(x_i, y_i)\}_{i=1}^n$, be a training set of size $n$ sampled *i.i.d.* from $\mu \times \eta$. Due to space limitation, proofs of all our results (organized section wise) are deferred to the Appendix.

For $x \in \mathcal{S}^{d-1}$, the `EaS` representation of $x$, that uses $k$-WTA sparsification step, is given by the function $h_1 : \mathcal{S}^{d-1} \to \{0, 1\}^m$ defined as,

$$h_1(x) = \Gamma_k(\Theta x), \tag{1}$$

where $\Theta$ is a $m \times d$ random projection matrix whose rows $\theta_1, \ldots, \theta_m$ are sampled i.i.d from the uniform distribution over $\mathcal{S}^{d-1}$ and $\Gamma_k : \mathbb{R}^m \to \{0, 1\}^m$ is the $k$-WTA function converting a vector in $\mathbb{R}^m$ to one in $\{0, 1\}^m$ by setting the largest $k \ll m$ elements of $\Theta x$ to 1 and the rest to zero. For any $j \in [m]$, let $C_j = \{x \in \mathcal{X} : h_1(x)[j] = 1\}$. We note that the subsets (response regions) $C_1, \ldots, C_m$ does not form a partition since they can be overlapping. We summarize our first algorithm for binary classification using $h_1$ in Alg. 1. During its learning/training phase, a vector $w \in [0, 1]^m$ summarizes the average $y$ value over $m$ response regions $C_j, j \in [m]$ using the training set. In particular, $w[j], j \in [m]$ learned during the training phase is precisely given by

$$\hat{\eta}_j = \frac{\sum_{i=1}^n y_i \mathbb{I}[x_i \in C_j]}{\sum_{i=1}^n \mathbb{I}[x_i \in C_j]} \tag{2}$$

Using (2), for any $x \in \mathcal{X}$, we further define

$$\hat{\eta}(x) = \frac{1}{k} \sum_{j:x \in C_j} \hat{\eta}_j \tag{3}$$

During the inference phase, for any test point $x$, Alg. 1 first computes $(w \cdot h_1(x))/k$, which is an average of average $y$ values over $k$ response regions $\{C_j : h_1(x)[j] = 1\}$, which can be interpreted as an estimate of the conditional probability $\eta(x)$ and is precisely the quantity $\hat{\eta}(x)$ given in (3). Alg. 1 then makes its prediction based on whether $\hat{\eta}(x)$ is greater than or equal to $1/2$. We can rewrite this conditional probability estimate as follows.

$$
\begin{aligned}
\hat{\eta}(x) &= \frac{w \cdot h_1(x)}{k} = \frac{1}{k} \sum_{j:x \in C_j} w[j] = \frac{1}{k} \sum_{j:x \in C_j} \hat{\eta}_j = \frac{1}{k} \sum_{j:x \in C_j} \frac{\sum_{i=1}^n y_i \mathbb{I}[x_i \in C_j]}{\sum_{i=1}^n \mathbb{I}[x_i \in C_j]} \\
&= \sum_{i=1}^n \underbrace{\left( \frac{1}{k} \sum_{j:x \in C_j} \frac{\mathbb{I}[x_i \in C_j]}{\sum_{i=1}^n \mathbb{I}[x_i \in C_j]} \right)}_{w_{n,i}(x)} y_i = \sum_{i=1}^n w_{n,i}(x) y_i
\end{aligned} \tag{4}
$$

Using viewpoint (4), Alg. 1 can be interpreted as a "plug-in" classifier Devroye et al. [1996] where prediction is based on whether the estimated conditional probability $\hat{\eta}$ exceeds $1/2$ or not. In particular, classifier in Alg. 1 can be represented as $g : \mathcal{X} \to \{0, 1\}$ described as

$$
g(x) = \begin{cases} 1, & \text{if } \hat{\eta}(x) \geq 1/2, \\ 0, & \text{otherwise.} \end{cases} \tag{5}
$$

In comparison, the Bayes optimal classifier $g_* : \mathcal{X} \to \{0, 1\}$ is defined as

$$
g_*(x) = \begin{cases} 1, & \text{if } \eta(x) \geq 1/2, \\ 0, & \text{otherwise.} \end{cases} \tag{6}
$$

In equation 4, the weights $w_{n,i}(x) = w_{n,i}(x, x_1, \ldots, x_n) \in \mathbb{R}$ depends on $x_1, \ldots, x_n$. Next we show that for sufficiently large $n$ sum of these weights is 1. Therefore, $\hat{\eta}$ is simply a weighted average estimator of $\eta$ and is a non-parametric classifier Devroye et al. [1996].

**Lemma 3.1.** *For any $x \in \mathcal{X}$, suppose $n$ is sufficiently large such that $\{x_i, \ldots, x_n\} \cap C_j \neq \emptyset$ for each $j$ satisfying $x \in C_j$. Then, $\sum_{i=1}^n w_{n,i}(x) = 1$.*

Indeed, we show in Lemma D.7 (in the Appendix), that for any $x \in \mathcal{X}$, whenever $n \to \infty$ and $m^k/n \to 0$ as $n \to \infty$, $|\{x_1, \ldots, x_n\} \cap C_j| \to \infty$ for all $j$ such that $x \in C_j$.

## 3.1 Consistency of Algorithm 1

In this section we prove that Alg. 1 is universally consistent. We start with the definition of consistent and universally consistent classification rule Devroye et al. [1996]. Let $D_n = \{(x_1, y_1), \ldots, ((x_n, y_n)\}$ be a training set sampled *i.i.d.* from a certain distribution of $(x, y)$ and let $g_n : \mathcal{X} \to \{0, 1\}$ be a classification rule learned using $D_n$. Then $g_n$ is consistent (or asymptotically Bayes-risk efficient) for a certain distribution of $(x, y)$ if the expected error probability $\mathbb{E}L_n = \Pr\{g_n(x, D_n) \neq y\} \to L^*$ as $n \to \infty$, where $L^*$ is the Bayes error probability. A sequence of decision rules is called universally consistent, if it is consistent for any distribution of the pair $(x, y)$. It is well known that a general theorem by Stone Stone [1977] (presented below) provides a recipe for establishing universal consistency of any classification rule of the form

$$
g_n(x) = \begin{cases} 1, & \text{if } \eta_n(x) \geq 1/2, \\ 0, & \text{otherwise.} \end{cases} \tag{7}
$$

based on an estimate $\eta_n(x)$ of the conditional probability $\eta(x)$, satisfying $\eta_n(x) = \sum_{i=1}^n w_{n,i}(x) y_i$ where weights $w_{n,i}(x) = w_{n,i}(x, x_1, \ldots, x_n)$ are non-negative and $\sum_{i=1}^n w_{n,i}(x) = 1$.

**Theorem 3.2.** *[Stone's Theorem (Theorem 6.3 [Devroye et al., 1996])] Assume that for any distribution of $x$, the weights satisfy the following three conditions:*
*(i) There is a constant $c$ such that, for every non-negative measurable function $f$ satisfying $\mathbb{E}f(x) < \infty$, $\mathbb{E}\left(\sum_{i=1}^n w_{n,i}(x)f(x_i)\right) \leq c\mathbb{E}f(x)$. (ii) For all $a > 0$, $\lim_{n\to\infty} \mathbb{E}\left(\sum_{i=1}^n w_{i,n}(x)\mathbb{I}_{\{\|x_i-x\|>a\}}\right) = 0$. (iii) $\lim_{n\to\infty}\mathbb{E}\left(\max_{1\leq i\leq n} w_{n,i}(x)\right) = 0$.*

*Then $g_n$ is universally consistent.*

Since we have already established in the previous section that the classification rule given in Alg. 1 admits the form given in (5), in conjunction with (4) and Lemma 3.1, we are ready to use Theorem 3.2 to establish universal consistency of Alg. 1. We state our consistency result below.

**Theorem 3.3.** *Let $(k_n)$ and $(m_n)$ be increasing functions of $n$ and let $(\delta_n)$ be a decreasing function of $n$ satisfying $\lim_{n\to\infty} \delta_n = 0$. Then the classification rule using $\mathsf{EaS}$ representation* (1) *given in Alg. 1 that admits the form given in* (5) *is universally consistent whenever the following conditions hold:(i) $\frac{k_n}{m_n} \to 0$ as $n \to \infty$, (ii) $\frac{\log(1/\delta_n)}{m_n} \to 0$ as $n \to \infty$ and, (iii) $\frac{m_n^{k_n}}{n} \to 0$ as $n \to \infty$.*

Once $m$ and $k$ are fixed, for any $x \in \mathcal{X}$, there are $\binom{m}{k}$ possible subsets of $k$ indices, from the $m$ possible indices in the $\mathsf{EaS}$ representation, that $h_1(x)$ can set to 1. For $i \in \{1, \ldots, \binom{m}{k}\}$, let $\sigma(i)$ represents the set of $k$ indices in the $i^{th}$ subset and we let $C_i^k = \{x \in \mathcal{X} : h_1(x)[j] = 1, \ \forall j \in \sigma(i)\}$ to denote the set of instances whose $\mathsf{EaS}$ representation precisely sets the indices in $\sigma(i)$ to 1 and the rest to zero. The following two results are crucial in proving our consistency results above, as well as in deriving convergence rate in the next section.

**Lemma 3.4.** *For any positive integers $m, k$, where $k < m$, the following relations hold.*

*i) For any $i \in \{1, \ldots, \binom{m}{k}\}$, $C_i^k = \cap_{j \in \sigma(i)} C_j$. ii) $\{C_i^k\}_{i=1}^{\binom{m}{k}}$ forms a partition of $\mathcal{X}$. iii) For any $j \in [m]$, $C_j = \cup_{i:j\in\sigma(i)} C_i^k$. iv) For any $j \in [m]$, $\mu(C_j) = \sum_{i:j\in\sigma(i)} \mu(C_i^k)$.*

**Lemma 3.5.** *Let $d \geq 3$ and pick any $0 < \delta < 1$. There is an absolute constant $c_0 > 0$ such that the following holds. With probability at least $1 - \delta$,*

$$\max_{j\in[m]} \operatorname{diam}(C_j) \leq 8 \left( \frac{k + c_0(d\log m + \log(1/\delta))}{m} \right)^{\frac{1}{d-1}}.$$

*In particular, if $k \geq c_0(d\log m + \log(1/\delta))$ then, $\max_{j\in[m]} \operatorname{diam}(C_j) \leq 8 \left( \frac{2k}{m} \right)^{\frac{1}{d-1}}$.*

## 3.2 Convergence rate of Algorithm 1

While the consistency result established in the previous section provides the behavior of Alg. 1 in the large sample limit, it does not provide the rate at which the excess Bayes risk converges to zero. In this section we establish such finite sample convergence rate. From the plug-in classifier view point of Alg. 1 given in (5) and the definition of Bayes optimal classifier in (6), it is easy to see that how well the prediction of Algorithm 1 relates to the prediction of the Bayes optimal classifier will depend on how well $\hat{\eta}(x)$ approximates $\eta(x)$.

Recall that for any $x \in \mathcal{X}$, point-wise error probability or risk of the Bayes optimal classifier is defined as $L^*(x) = \Pr(g^*(x) \neq y|x) = \min\{\eta(x), 1 - \eta(x)\}$ [Devroye et al., 1996]. Taking expectation over $x$, risk of the Bayes optimal classifier is $L^* = \mathbb{E}(\min\{\eta(x), 1 - \eta(x)\})$. Now, for any plug-in classification rule $g_n(x)$ defined in (7) that uses a conditional probability estimate $\eta_n$ using training data $D_n$, the excess Bayes risk is given by the following well known result (Corollary 6.1 of Devroye et al. [1996].

**Lemma 3.6** ([Devroye et al., 1996])**.** *The error probability of a classifier $g_n(x)$ defined in* (7) *satisfies the inequality:* $\Pr(g_n(x) \neq y|D_n) - L^* \leq 2\mathbb{E}_x\left( |\eta(x) - \eta_n(x)| \Big| D_n \right).$

Taking expectation over $D_n$, the above lemma immediately translates to

$$\Pr(g_n(x) \neq y) - L^* \leq 2\mathbb{E}_{x,D_n}\left( |\eta(x) - \eta_n(x)| \right) \tag{8}$$

Therefore, in order to establish convergence rate of Alg. 1, following Lemma 3.6 and (8), we need to bound expected value of $|\hat{\eta}(x) - \eta(x)|$. Towards this end, we first extend $\eta$ defined over points $x \in \mathcal{X}$ to over measurable sets $\mathcal{A} \subset \mathcal{X}$ with $\mu(\mathcal{A}) > 0$ as follows:

$$\eta(\mathcal{A}) = \frac{1}{\mu(\mathcal{A})} \int_{\mathcal{A}} \eta(x)\mu(dx) = \frac{1}{\mu(\mathcal{A})} \int_{\mathcal{A}} \Pr(y=1|x)\mu(dx) = \Pr(y=1|x \in \mathcal{A}) = \mathbb{E}(y|x \in \mathcal{A}) \tag{9}$$

Thus, $\eta(\mathcal{A})$ is the probability that $y = 1$ for a point $x$ chosen at random from the distribution $\mu$ restricted to the set $\mathcal{A}$, or in other words, it is the expected value of $y$ when $x$ is restricted to the set $\mathcal{A}$.

With this definition, we introduce an intermediate quantity

$$\bar{\eta}(x) = \frac{1}{k} \sum_{j:x \in C_j} \bar{\eta}_j \tag{10}$$

where, $\bar{\eta}_j$ is defined as,

$$\bar{\eta}_j = \frac{1}{\mu(C_j)} \int_{C_j} \eta(x)\mu(dx) = \eta(C_j) \tag{11}$$

Using triangle inequality, this allows us to write: $\mathbb{E}_{x,D_n}|\hat{\eta}(x) - \eta(x)| = \mathbb{E}_{x,D_n}|\hat{\eta}(x) - \bar{\eta}(x) + \bar{\eta}(x) - \eta(X)| \leq \mathbb{E}_{x,D_n}|\hat{\eta}(x) - \bar{\eta}(x)| + \mathbb{E}_{x,D_n}|\bar{\eta}(x) - \eta(x)|$, and we show how to individually bound each term on the right-hand side of this inequality next.

*Remark* 3.7. Note that, once $m$ and $k$ are fixed, our hypothesis space is the set of all linear models on the $k$-sparse $m$ dimensional binary vectors. The two terms on the right-hand side of the above inequality correspond to *estimation error* (the error of our proposed classifier with respect to the best hypothesis from the hypothesis space) and *approximation error* (the error difference between the best hypothesis from the hypothesis space and the target classifier, i.e., Bayes optimal), respectively.

## 3.3 Bounding $\mathbb{E}_{x,D_n}|\hat{\eta}(x) - \bar{\eta}(x)|$

Note that there is an inherent randomness in our proposed algorithm associated with the choice of $\Theta$. In particular, the response regions $C_j$ are random quantities that depend on the choice of $\Theta$. In this section, we fix $\Theta$ and conditioned on this, we bound $\mathbb{E}_{x,D_n}|\hat{\eta}(x) - \bar{\eta}(x)|$. This ensures that for any $\delta > 0$, the same bound holds with probability at least $1 - \delta$ over the choice of $\Theta$.

**Lemma 3.8.** *Fix any* $\Theta$. *Then we have,* $\mathbb{E}_{x,D_n}|\hat{\eta}(x) - \bar{\eta}(x)| \leq \sqrt{\frac{m}{kn}}$.

***Sketch of proof:*** From the definition of $\hat{\eta}(x)$ and $\bar{\eta}(x)$ given in (3) and (10) and applying Jensen's inequality, crux of the proof is to focus on the expected value of the random quantity $\frac{1}{k}\sum_{j:x \in C_j}\left(\frac{\sum_{i=1}^n y_i \mathbb{I}[x_i \in C_j]}{\sum_{i=1}^n \mathbb{I}[x_i \in C_j]} - \eta(C_j)\right)^2$. Note that, for $j \in [m]$, $\sum_{i=1}^n \mathbb{I}[x_i \in C_j] = n\mu_n(C_j)$ is the number of the points from $D_n$ that fall in $C_j$, where $\mu_n(C_j)$ is the empirical probability estimate. We bound the quantity of interest in Lemma 3.8 as a sum of two quantities corresponding to the following two cases:

**Case 1: when $\mu_n(C_j) = 0$.** Using the notation $0/0 = 0$, the quantity of interest becomes $\frac{1}{k}\sum_{j:x \in C_j}\left(\eta^2(C_j)\mathbb{I}[\mu_n(C_j) = 0]\right)$. Utilizing the properties of the response regions established in Lemma 3.4, we show in Lemma E.3 that the expected value of the quantity of interest at most $\frac{m}{nke}$.

**Case 2: when $n\mu_n(C_j) > 0$.** Here the quantity of interest becomes $\frac{1}{k}\sum_{j:x \in C_j}\left(\frac{\sum_{i=1}^n (y_i - \eta(C_j))\mathbb{I}[x_i \in C_j]}{n\mu_n(C_j)}\right)^2\mathbb{I}[n\mu_n(C_j) > 0]$. Conditioned on $x, x_1, \ldots, x_n$, we first show in Lemma E.1 that expected value (w.r.t. $y_1, \ldots, y_n$) of this quantity becomes at most $\frac{1}{4k}\sum_{j:x \in C_j}\left(\frac{\mathbb{I}[n\mu_n(C_j) > 0]}{n\mu_n(C_j)}\right)$. Next, conditioned on $x_1, \ldots, x_n$, and utilizing the properties of the response regions established in Lemma 3.4, we show in Lemma E.2 that expected value (w.r.t. $x$) of this quantity is at most $\frac{1}{4}\sum_{j=1}^m\left(\frac{\mu(C_j)\mathbb{I}[n\mu_n(C_j) > 0]}{n\mu_n(C_j)}\right)$. Finally, using standard Binomial bound (Lemma E.4) we show that expected value (w.r.t. $x_1, \ldots, x_n$) of this quantity is at most $\frac{m}{2kn}$.

## 3.4 Bounding $\mathbb{E}_{x,D_n}|\bar{\eta}(X) - \eta(x)|$

In order to bound the expected value of $|\bar{\eta}(x) - \eta(x)|$, we need to impose certain smoothness condition on $\eta$. We consider a general form of smoothness, known as Hölder continuty for $\eta$.

**Definition 3.9.** We say that $\eta : \mathcal{X} \to [0,1]$ is $(L, \beta)$ smooth if for all $x, x' \in \mathcal{X}$, we have $|\eta(x) - \eta(x')| \leq L\|x - x'\|^\beta$.

Using Hölder continuity assumption above, we first show the following:

**Lemma 3.10.** *Suppose* $\eta$ *is* $(L, \beta)$ *smooth. Then,* $\sup_{x \in \mathcal{S}}|\eta(x) - \bar{\eta}(x)| \leq L \cdot \max_{j \in [m]}(\text{diam}(C_j))^\beta$.

We have already shown in Lemma 3.5 how to bound the diameters of $C_j$. Combining these two results we have

**Lemma 3.11.** *Let $d \geq 3$ and pick any $0 < \delta < 1$. Assume $\eta$ to be $(L, \beta)$ smooth. There is an absolute constant $c_0 > 0$ such that the following holds. If $k \geq c_0(d \log m + \log(1/\delta))$ then with probability at least $1 - \delta$, $\mathbb{E}_{x,D_n}|\bar{\eta}(x) - \eta(x)| \leq 8L \left(2k/m\right)^{\frac{\beta}{d-1}}$.*

Combining Lemma 3.8, 3.11 and 3.6 we are present the main result of this section.

**Theorem 3.12.** *Let $D_n = \{(x_i, y_i)\}_{i=1}^n \subset \mathcal{X} \times \{0, 1\}$ be the training data and consider the `EaS` representation given in (1). Let $d \geq 3$ and pick any $0 < \delta < 1$. Assume $\eta$ to be $(L, \beta)$ smooth. There is an absolute constant $c_0 > 0$ such that the following holds. If $k \geq c_0(d \log m + \log(1/\delta))$ then with probability at least $1 - \delta$ over the random choice of $\Theta$,*

$$\Pr(g(x) \neq y) - L^* \leq 2 \left( \sqrt{\frac{m}{kn}} + 8L \left(\frac{2k}{m}\right)^{\frac{\beta}{d-1}} \right)$$

**Corollary 3.13.** *In Theorem 3.12, set $m = kn^{\frac{(d-1)}{2\beta+(d-1)}}$. Then, with probability at least $1 - \delta$, over the choice of $\Theta$,*

$$\Pr(g(x) \neq y) - L^* = O \left( n^{-\frac{\beta}{2\beta+(d-1)}} \right).$$

*Remark* 3.14. Since $\mathcal{X} \subset \mathcal{S}^{d-1}$, the effective dimension in our setting is $d' = (d-1)$ and the convergence rate of Corollary 3.13 can be rewritten as $O \left( n^{-\frac{\beta}{2\beta+d'}} \right)$ which is minimax-optimal for plug-in classifiers under the assumption that $\eta$ is $(L, \beta)$-smooth [Audibert and Tsybakov, 2007].

### 3.5 Inability to adapt to manifold structure

In Theorem 3.12, we derived the convergence rate of the classifier presented in Alg. 1 by bounding $\mathbb{E}_{x,D_n}|\eta(x) - \hat{\eta}(x)|$, which upper bounds the excess Bayes risk, from above and the resulting convergence rate decays exponentially slowly with the dimension $d$. We now show that even if the data lie on a low dimensional manifold having dimesnion $d_0 \ll d$, there exists a smooth $\eta$ such that the quantity $\mathbb{E}_{x,D_n}|\eta(x) - \hat{\eta}(x)|$ decreases at a rate no faster than $n^{-\frac{1}{d+1}}$. To prove claim, we assume data to lie on the following one-dimensional manifold:

$$\mathcal{X}_1 = \{(x_1, x_2, 0, 0, \ldots, 0) \in \mathbb{R}^d : x_1^2 + x_2^2 = 1\} \tag{12}$$

We further assume that $k = \beta = 1$ which implies that $\eta$ is $L$-Lipschitz from the definition of $(L, \beta)$-smoothness. Our lower bound result is as follows.

**Theorem 3.15.** *For any $d > 3$, let input space $\mathcal{X}_1$ be the one-dimensional sub-manifold of $\mathbb{R}^d$ given in (12). Take $k = 1$ and $\beta = 1$. Suppose the random matrix $\Theta$ has rows chosen from a distribution that is uniform over $\mathcal{S}^{d-1}$. Then there exists a $\frac{1}{2}$-Lipschitz function $\eta : \mathcal{X}_1 \to [0, 1]$ such that the following holds with probability at least $1/2$ over the choice of $\Theta$.*

$$\mathbb{E}_{x,D_n}|\eta(x) - \hat{\eta}(x)| = \Omega \left( n^{-\frac{1}{d+1}} \right)$$

## 4 Algorithm 2

In this section we present an alternate `EaS` representation and an associated classification algorithm that adapts to intrinsic dimension $d_0$ when data lie on a low dimensional manifold with dimension $d_0 \ll d$. Here, we assume that the rows $\theta_i, i \in [m]$ of matrix $\Theta$ are sampled *i.i.d.* from a multivariate Gaussian distribution $N(0, 1/\sqrt{d}I_d)$, denote by $\nu$, where $I_d$ is $d \times d$ identity matrix. This `EaS` representation is denoted by $h_2 : \mathcal{X} \to \{0, 1\}^m$, where for any point $x \in \mathcal{X}$, the $j^{th}$ coordinate of $h_2(x)$ is set to 1, as given in (14), using a data dependent threshold $\tau_n$. In particular, given a training set of size[1] $2n$ sampled *i.i.d.* from $(\mu, \eta)$, using the first half of it, namely, $D'_n = \{(x'_1, y'_1), \ldots, (x'_n, y'_n)\}$, define $\tau_n : \mathbb{R}^d \to \mathbb{R}$ to be:

$$\tau_n(\theta) = \sup\left\{\tau : \frac{1}{n} \sum_{i=1}^n \mathbb{I}[\theta \cdot x'_i \geq \tau] \geq \frac{k}{m}\right\} \tag{13}$$

---

[1]Instead of representing the training set to be $D_{2n} = \{(x_i, y_i)\}_{i=1}^{2n}$, for notational convenience, we represent its first half as $D'_n = \{(x'_i, y'_i)\}_{i=1}^n$ and the second half as $D_n = \{(x_i, y_i)\}_{i=1}^n$. $D'_n$ is only used to compute empirical thresholds in (13) and therefore, we could use unlabeled data sampled *i.i.d.* from $\mu$ for this purpose.

$$h_2(x)[j] = \mathbb{I}[\theta_j \cdot x \geq \tau_n(\theta_j)] \tag{14}$$

We call the sparsification scheme given in (14) *empirical-k-thresholding*. For $j \in [m]$, the $j^{th}$ response region $C_j$ is defined as:

$$C_j = C(\theta_j) = \{x \in \mathcal{X} : \theta_j \cdot x \geq \tau_n(\theta_j)\} \tag{15}$$

Using the second half of the training data, namely $D_n = \{(x_1, y_1), \ldots, (x_n, y_n)\}$, average $y$ values over different $C_j$ are estimated and summarized in vector $w$ in the same way as in Alg. 1 implying that $w[j] = \hat{\eta}_j, \forall j \in [m]$, where $\hat{\eta}_j$ is given in (2). This completes the training phase of our proposed algorithm summarized in Alg. 2.

---

**Algorithm 2** Training set $D'_n = \{(x'_i, y'_i)\}_{i=1}^n, D_n = \{(x_i, y_i)\}_{i=1}^n \subset \mathcal{X} \times \{0, 1\}$, Projection dimensionality $m \in \mathbb{N}$, integer $k \ll m$, integer $t$, random seed $R$, and inference with test point $x \in \mathcal{S}^{d-1}$.

---

**Train**EaSClassifier$(D_n, D'_n m, k, t, R)$
   Sample $\Theta$ with seed $R$
   Initialize $w[i], \mathtt{ct}[i] \leftarrow 0, \forall i \in [m]$
   **for** $j \in [m]$ **do**
     | Compute $\tau_n(\theta_j)$ using (13) and $D'_n$
   **end**
   **for** $(x, y) \in D_n$ **do**
     $\mathtt{eas} \leftarrow h_2(x)$
     $w[i] \leftarrow w[i] + y, \forall i \in [m] : \mathtt{eas}[i] = 1$
     $\mathtt{ct}[i] \leftarrow \mathtt{ct}[i]+1, , \forall i \in [m] : \mathtt{eas}[i] = 1$
   **end**
   $w[i] \leftarrow w[i]/\mathtt{ct}[i], \forall i \in [m]$
   **return** $\Theta, w$
**end**
**Infer**EaSClassifier$(x, \Theta, t, w)$
   $\mathtt{eas} \leftarrow h_2^n(x)$
   $\Theta_x \leftarrow \{\theta_i : h_2(x)[i] = 1\}$
   $\mathtt{e\tilde{a}s} \leftarrow \mathtt{eas}$
   $\mathtt{e\tilde{a}s}[i] \leftarrow 0, \forall i \in [m] : i \notin A_t(x)$
   **return** $\mathbb{I}[(\mathtt{e\tilde{a}s} \cdot w)/t \geq \frac{1}{2}]$
**end**

---

The inference phase of Alg. 2 is slightly different from Alg. 1. While EaS representation using $h_2$ is not $k$-sparse anymore, we show that for large enough sample size, with high probability, it is at least $k/2$-sparse and at most $2k$-sparse in expectation (see Lemma F.1 in Appendix F). Let $t$ be an integer passed as an argument to Alg. 2. For any $x \in \mathcal{X}$, let $\Theta(x) = \{\theta_j : x \in C_j\}$ and we define $A_t(x)$ to be the set containing the indices $j \in [m]$, such that $\theta_j$ is one of the $t$ closest points to $x$ from $\Theta(x)$. To make an inference for any $x \in \mathcal{X}$, Alg. 2 first computes $(\mathtt{e\tilde{a}s} \cdot w)/t$ and makes it prediction based on whether this quantity is greater than $1/2$. Clearly, $(\mathtt{e\tilde{a}s} \cdot w)/t$ is the average of $w[j]$ for $j \in A_t(x)$ and therefore, the conditional probability estimate $\hat{\eta}(x)$ of Alg. 2 can be represented as,

$$\hat{\eta}(x) = \frac{1}{t} \sum_j \hat{\eta}_j \mathbb{I}[j \in A_t(x)] \tag{16}$$

*Remark* 4.1. Note that $h_1$ and $h_2$ are different in a specific way. When the support of data does not cover the whole unit sphere and is concentrated possibly in a small region and $m$ is large, many of the $m$ coordinates in EaS representation will never be activated (set to 1) for any data point as the corresponding projection direction $\theta_j$ may not be one of the $k$ closest ones to any data point. Thus, many of the $\theta_j$ will be unused. This problem is avoided in $h_2$, where every $\theta_j$ is used but the respective response region $C_j$ may not be local to the manifold. For this purpose we need to identify "good" projection directions. Later in Lemma B.4 we show that, for any $\delta > 0$, with probability at least $1 - 2\delta$ over the choice of $\Theta$ and $D'_n$, the number of "good" projection directions $t$ is linear in $k$.

Due to space limitation, manifold assumptions and other important details of analysis of Alg. 2 are presented in Appendix B and we present the main theoretical results below.

**Theorem 4.2.** *Let $D_n = \{(x_i, y_i)\}_{i=1}^n \cup D'_n = \{(x'_i, y'_i)\}_{i=1}^n \subset \mathcal{X} \times \{0, 1\}$ be the training data where the data lies on a low dimensional manifold satisfying manifold assumption presented in section B.1 and suppose the EaS representation is given as in (14). Pick any $0 < \delta < 1$. Assume $\eta$ to be $(L, \beta)$ smooth. If $k \geq c'_d \ln(m/\delta)$, where $c'_d$ is a constant that depend on $d$, then with probability at least $1 - 2\delta$,*

$$\Pr(g(x) \neq y) - L^* \leq 2 \left( \sqrt{\frac{2m}{\alpha_d kn}} + 4L \left(\frac{2k}{c_1 m}\right)^{\frac{\beta}{d_0}} \right)$$

*where $\alpha_d$ is a constant that depends on $d$.*

**Corollary 4.3.** *In Theorem 4.2, setting $m = kn^{\frac{d_0}{2\beta+d_0}}$ ensure that with probability at least $1 - \delta$,*

$$\Pr(g(x) \neq y) - L^* = O\left(n^{-\frac{\beta}{2\beta+d_0}}\right).$$

*Remark* 4.4. The convergence rate of Corollary 4.3 depends only on $d_0$ and is minimax-optimal for plug-in classifiers under the assumption that $\eta$ is $(L, \beta)$-smooth [Audibert and Tsybakov, 2007].

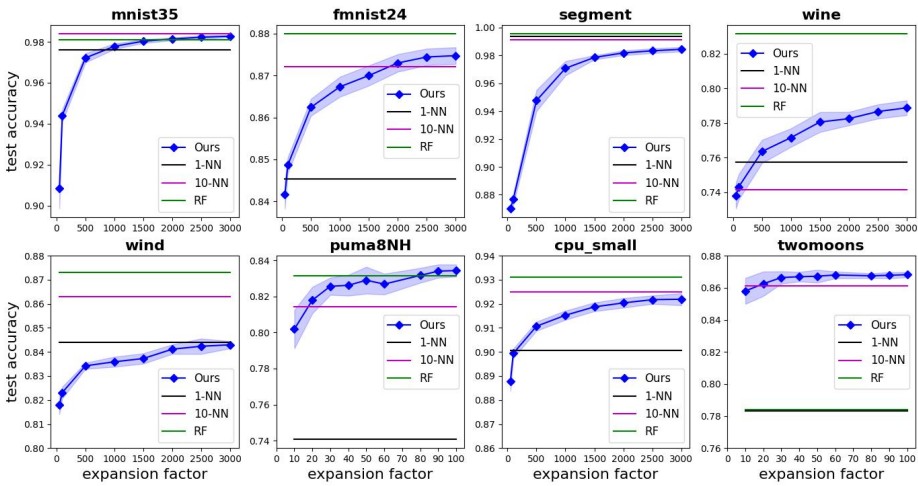

Figure 2: Empirical evaluation of Alg. 1, $k$-NN (for $k = 1$ and 10) and RF on eight datasets Here expansion factor is $m/d$. An error bar in the form of a shaded graph is provided for Alg. 1 over 10 independent runs.

## 5  Empirical evaluations

We investigate the effectiveness of our proposed method by evaluating it on eight benchmark datasets, details of which are provided in appendix A. We address the following questions:
1. Does performance our proposed classifier improve with increasing $m$ as suggested by the theory?
2. How does our proposed classifier perform compared to other non-parametric classifiers?
For each dataset, we generate train and test set using scikit-learn's train_test_split method (80 : 20 split). We compare our proposed method against two non-parametric classifiers – scikit-learn's implementation of $k$-nearest neighbor classifier ($k$-NN) and random forest (RF). For $k$-NN, we used two values of $k$: $k = 1$ and 10. For RF we use a grid search over the number of estimators (trees) from the set $\{250, 500, 750, 1000\}$ and perform a 3 fold cross validation to choose the final model. We preset our experimental results in Fig. 2 where we plot test accuracy of Alg. 1, $k$-NN (for $k = 1$ and 10) and RF by varying expansion factor, where we define expansion factor to be $m/d$. As per Theorem 3.12, we set $k$ to be $d \log m$. As can be seen from Fig. 2, with increasing $m$ test accuracy of Alg. 1 increases in all eight datasets and becomes comparable to that of $k$-NN and RF for large $m$, thus corroborating our theoretical findings.

## 6  Conclusions, limitations and future work

In this paper, we present an interesting connection between non-parametric estimation and expansion-and-sparsify representation. We presented two non-parametric classification algorithms using EaS representation and proved that both algorithms yield minimax-optimal convergence rates. The convergence rate of the first algorithm depends on the ambient dimension $d$, while the convergence rate of the second algorithm, under manifold assumption, depends only on the intrinsic dimension $d_0 \ll d$. In both algorithms, the projection directions are chosen in a data-independent manner. One limitation of our current work is that, even though the second algorithm adapts to the manifold structure, there is a large constant, possibly depending exponentially on $d$, involved in bounding the excess Bayes risk, that is hidden under the Big Oh notation. In the future, we plan to investigate various data-dependent projection direction choices for a sparse representation, that would adapt to a manifold structure, and the constant involved in bounding of the excess Bayes risk from above, would be independent of ambient dimension $d$.

**Acknowledgements:** We thank the anonymous reviewers for their constructive feedback. This work is supported by funding from the "NSF AI Institute for Foundations of Machine Learning (IFML)" (FAIN:2019844).

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

| Dataset Name | # Samples | # Features |
|---|---|---|
| mnist35 | 13454 | 20 |
| fmnist24 | 11200 | 20 |
| segment (v.2) | 2310 | 19 |
| wine (v.7) | 2554 | 11 |
| wind (v.2) | 6574 | 14 |
| puma8NH (v.2) | 8192 | 8 |
| cpu_small (v.3) | 8192 | 12 |
| twomoons | 5000 | 2 |

Table 1: Dataset statistics

## A  Dataset details and computing environment

Details of the eight datasets used in our experiments are listed in Table 1. Among the eight datasets, the `twomoons` is a synthetic dataset generated using `make_moons` method from `sklearn.dataset`[2] with noise parameter $0.2$. All the remaining datasets are taken from the OpenML repository[3] Vanschoren et al. [2013]. Both `mnist` and `fmnist` (`fashion-mnist` for short) are 10 class classification problem with 784 features. We convert them to binary classification problems by using the label 3 and 5 for the `mnist` dataset and label 2 (Pullover) and 4 (Coat) for `fmnist` dataset. For efficiency purpose, the feature dimensions for both these datasets are reduced to 20 using principal component analysis (PCA). For the remaining six datasets, the task is that of binary classification. For all eight datasets, the features are normalized using `StandardScaler` option in `scikit-learn` and are made to be unit norm.

We run our experiments on a laptop with Intel Xeon W-10855M Processor, 64GB memory and NVIDIA Quadro RTX 5000 Mobile GPU (with 16GB memory).

## B  Missing details of convergence rate of Algorithm 2 from section 4

In this section we present missing details of convergence rate analysis of Algorithm 2 from section 4. Proofs of various technical results presented in this section as well as the proof of Theorem 4.2 are deferred to section B.

### B.1  Manifold assumption

To analyze convergence rate of Algorithm 2, we proceed in the same was as in section 3.2. However, to take advantage of the fact that data lie on a low-dimensional manifold in our analysis, we make similar manifold assumptions as in Dasgupta and Tosh [2020]. The input space $\mathcal{X}$ is a compact $d_0$-dimensional Riemannian sub manifold $M$ of $\mathbb{R}^d$ contained in the unit sphere, that is $M \subset \mathcal{S}^{d-1}$. $M$ has nice boundaries and that the distribution on it $\mu$, is almost uniform: formally, there exists constants $c_1, c_2, c_3 > 0$ such that for all $x \in M$ and for all $r \leq r_0$ (where $r_0$ is an absolute constant)

$$c_1 r^{d_0} \leq \mu(B_M(x, r)) < c_2 r^{d_0} \tag{17}$$

$$\text{Vol}(B_M(x, r)) \geq c_3 r^{d_0} \tag{18}$$

Here $B_M(x, r) = B(x, r) \cap M$.

To effectively analyze the data distribution supported on a manifold, we impose conditions on the curvature by adopting the common requirement that $M$ has a positive reach $\rho > 0$: that is, every point in an open tubular neighborhood of $M$ of of radius $\rho$ has a unique nearest neighbor in $M$ [Niyogi et al., 2008]. For each $x \in M$, let $N(x)$ denotes the $(d - d_0)$ dimensional subspace of normal vectors to the tangent plane at $x$. For each $x \in M$, the sets $\Gamma_\rho(x) = \{x + ru \in \mathbb{R}^d : u \in N(x), \|u\| = 1, 0 < r < \rho\}$ are disjoint and let $\tau_\rho$ be their union. Let $\pi_M : \tau_\rho \to M$ be the projection map that sends any point in $\tau_\rho(x)$ to $x$, its nearest neighbor in $M$.

---

[2]https://scikit-learn.org/stable/
[3]https://www.openml.org/

## B.2 Bounding $\mathbb{E}_{x,D_n}|\hat\eta(x) - \bar\eta(x)|$

The response regions $C_j, j \in [m]$ are now random quantities that depend on the choice of $\Theta$ as well as $D'_n$ used to compute individual thresholds. The quantity $\hat\eta(x)$ given in (16) depends on $t$. In this section, we fix $\Theta$ and $t$ and conditioned on this, we bound $\mathbb{E}_{x,D_n,D'_n}|\hat\eta(x) - \bar\eta(x)|$. Later in Lemma B.4 we show that, for any $\delta > 0$, with probability at least $1 - 2\delta$ over the choice of $\Theta$ and $D'_n$, $t$ is linear in $k$.

**Lemma B.1.** *Fix any $\Theta$ and $t$. Then the following holds.*

$$\mathbb{E}_{x,D_n,D'_n}|\hat\eta(x) - \bar\eta(x)| \leq \sqrt{\frac{m}{tn}}.$$

***Sketch of proof:*** The proof strategy is similar to Lemma 3.8, but requires careful analysis to accommodate the facts that $C_j, j \in [m]$ depend on the choice of $D'_n$ and $\hat\eta$ in (16) has an indicator function. Details are provided in Appendix B.

## B.3 Bounding $\mathbb{E}_{x,D_n}|\bar\eta(x) - \eta(x)|$

Using identical proof strategy of Lemma 3.10, we have,

**Lemma B.2.** *Under empirical $k$-thresholding scheme, if $\eta$ is $(L, \beta)$ smooth, then for all $x \in C_1 \cup \cdots, \cup C_m$,*

$$|\eta(x) - \bar\eta(x)| \leq L \max_{j \in [m]}(\mathrm{diam}(C_j))^\beta$$

Unfortunately, unlike before, all $C_j$ such that $x \in C_j$ may not be local to $x$ and thus average $y$ values in some of these $C_j$, may be very different from $\eta(x)$. To address this, we call a $\theta \in \mathbb{R}^d$ *good* if it lies in $\Gamma_{\rho/2}$. Good $\theta_j$'s are close to the manifold $M$ and are guaranteed to be activated by a single neighborhood of $M$. Thus, for large enough $m$, if the diameters of the $C_j$'s corresponding to good $\theta_j$'s are made reasonable small then the average $y$ values in such $C_j$'s will be a reasonably close of $\eta(x)$ for $x \in C_j$. First, for any good $\theta_j$, we bound the diameter of $C_j$ emphasizing that the response region of a good $\theta_j$ is local.

**Lemma B.3.** *Pick any good $\theta \in \mathbb{R}^d$. Define $\Delta = \|\theta - \pi_M(\theta)\|$ to be the distance from $\theta$ to its projection on $M$. Let $C(\theta)$ be the response region associated with $\theta$ as defined in (15). Pick any $0 < \delta < 1$. Then with probability at least $1 - \delta$, over the choice of $D'_n$, the following holds:*

$$B_M\left(\pi_M(\theta), \sqrt{\frac{\rho - \Delta}{\rho + \Delta}\left(\frac{k}{2c_2 m}\right)^{1/d_0}}\right) \subset C(\theta) \subset B\left(\pi_M(\theta), \sqrt{\frac{\rho + \Delta}{\rho - \Delta}\left(\frac{2k}{c_1 m}\right)^{1/d_0}}\right)$$

*In particular this implies,* $\mathrm{diam}(C(\theta)) \leq 4\left(\frac{2k}{c_1 m}\right)^{1/d_0}$, *provided* $(2k/(c_1 m))^{1/d_0} < \min(\rho, r_0)$ *and $n$ satisfies $n \geq \frac{c_0 m}{k}\left(\log n + \log\left(\frac{m}{\delta}\right)\right)$, where $c_0 > 0$ is a universal constant.*

Next we show that each $x \in M$ has number of good $\theta_j$ linear in $k$ with high probability.

**Lemma B.4.** *Pick $\theta_1, \ldots, \theta_m \sim \nu$. There is a constant $c'_d$, depending on $d$, for which the following holds. Pick any $0 < \delta < 1$. Set $k \geq c'_d \ln(m/\delta)$. then with probability at least $1 - 2\delta$ over the choice of $\theta_j$'s and $D'_n$, for every $x \in M$, there are $\alpha_d k/2$ good $\theta_j$'s with $x \in C(\theta_j)$ where $c'_d$ and $\alpha_d$ are constants that depend on dimension $d$.*

Combining[4] Lemma B.3 and B.4, we have

**Lemma B.5.** *Suppose the data distribution is supported on a $d_0$-dimensional submanifold $M$ of $\mathbb{R}^d$ with reach $\rho > 0$, that additionally satisfied (17) and (18). Suppose that the rows of $\Theta$ are chosen from $N(0, I_d)$. There is a constant $c'_d$, depending on the dimension $d$, and $c_0 > 0$, universal constant, for which the following holds. Pick any $0 < \delta < 1$. Let $k, m$ and $n$ be chosen so that*

---

[4]We note that Lemma B.3 a modified version of Lemma 6 of Dasgupta and Tosh [2020] to incorporate the uncertainty associated with the data dependent threshold selection in (13). Also, Lemma B.4 is a modified version of Lemma 7 of Dasgupta and Tosh [2020] which incorporates threshold selection uncertainty and additionally guaranteeing the number of good $\theta_j$ to be linear in $k$.

$k \geq c'_d \ln(m/delta)$, $(2k/(c_1 m))^{1/d_0} < \min(\rho, r_0)$ *and* $n \geq (c_0 m/k)(\log n + \log(m/\delta))$. *Then with probability at least* $1 - 2\delta$ *over the choice of* $\Theta$ *and* $D'_n$,

$$\sup_{x \in \mathcal{X}} |\bar{\eta}(x) - \eta(x)| \leq 4L \left( \frac{2k}{c_1 m} \right)^{\frac{\beta}{d_0}}$$

Combining everything leads to the main result presented in Theorem 4.2.

# C  Various proofs from section 3

## C.1  Proof of Lemma 3.1

*Proof.* Simply interchange the summation. That is,

$$\sum_{i=1}^{n} w_{i,n}(x) = \sum_{i=1}^{n} \left( \frac{1}{k} \sum_{j: x \in C_j} \frac{\mathbb{I}[x_i \in \mathcal{C}_j]}{\sum_{i=1}^{n} \mathbb{I}[x_i \in \mathcal{C}_j]} \right) = \frac{1}{k} \sum_{j: x \in C_j} \left( \frac{\sum_{i=1}^{n} \mathbb{I}[x_i \in \mathcal{C}_j]}{\sum_{i=1}^{n} \mathbb{I}[x_i \in \mathcal{C}_j]} \right) = 1$$

$\square$

# D  Various proofs from section 3.1

In this section we present various technical results that are needed to prove our main theorem (Theorem 3.3). Due to space limitation proofs of some of these technical results are deferred to the supplementary material.

## D.1  Proof of Lemma 3.4

*Proof.* For part (i), note that if $x \in C_i^k$, then by definition, $\forall j \in \sigma(i), h_{\Theta,k}(x)[j] = 1$ which implies $x \in C_j$. Therefore, $C_i^k \subseteq \cap_{j \in \sigma(i)} C_j$. On the other hand, if $x \in \cap_{j \in \sigma(i)} C_j$ then $h_{\Theta,k}(x)[j] = 1 \ \forall j \in \sigma(i)$, which implies $x \in C_i^k$. Therefore, $\cap_{j \in \sigma(i)} C_j \subseteq C_i^k$.

For part (ii), take any $i, j \in [\binom{m}{k}], i \neq j$, then $\sigma(i) \neq \sigma(j)$. Therefore, if $x \in C_i^k$ then $x \notin C_j^k$. Since $x$ was arbitrary, $C_i^k \cap C_j^k = \emptyset$. Now, for any $x \in \mathcal{X}$, since $h_{\Theta,k}(x)$ has exactly $k$ bits activated (set to 1), there must exist $l \in [\binom{m}{k}]$ such that $x \in C_l^k$. Therefore, $\cup_{i=1}^{\binom{m}{k}} C_i^k = \mathcal{X}$.

For part (iii), take any $x \in C_j$. From the definition of $C_j$, $h_{\Theta,k}(x)[j] = 1$ and therefore, $x \in C_i^k$ for any $i$ such that $j \in \sigma(i)$. From part(ii) since such $C_i^k$s are disjoint and $x$ was arbitrary, the result follows.

Finally, part (iv) follows immediately from part (ii) and (iii) since $C_i^k$s are disjoint. $\square$

## D.2  proof of Lemma 3.5

*Proof.* While general idea of this result appeared in Dasgupta and Tosh [2020], we provide a simplified proof. Let $\nu(r) = \inf_{x \in \mathcal{X}} \nu(B(x, r))$. We first claim that if $\nu(r) \geq \frac{2}{m} \left( k + c_0 \left( d \log m + \log \left( \frac{1}{\delta} \right) \right) \right)$ then for every $j \in [m], C_j \subset B(\theta_j, r)$. By definition, every ball of radius $r$, centered at some $x \in \mathcal{X}$, has $\nu$-mass at least $\nu(r)$. Thus, by Lemma D.1, with probability at least $1 - \delta$, every $x \in \mathcal{X}$ will have its nearest $k$ $\theta_j$'s within a distance $r$. Therefore, the $j^{th}$ bit of the EaS representation given in (1) will be activated by points $x$ within distance $r$ of $\theta_j$. Therefore, $\text{diam}(C_j) \leq 2r$ for all $j = 1, \ldots, m$, where $r$ satisfies

$$\nu(B(x, r)) \geq \frac{2 \left( k + c_0 \left( d \log m + \log \left( \frac{1}{\delta} \right) \right) \right)}{m}$$

By Lemma D.2, we can take $r$ to be the value

$$r = \left(\frac{3\sqrt{d}}{(3/4)^{(d-1)/2}} \cdot \frac{2\left(k + c_0\left(d\log m + \log\left(\frac{1}{\delta}\right)\right)\right)}{m}\right)^{1/(d-1)}$$

$$= \frac{2}{\sqrt{3}}\left(\frac{6\sqrt{d}\left(k + c_0\left(d\log m + \log\left(\frac{1}{\delta}\right)\right)\right)}{m}\right)^{1/(d-1)}.$$

Therefore, we have

$$\text{diam}(C_j) \quad \le \quad 2r \le \frac{4}{\sqrt{3}}\left(\frac{6\sqrt{d}\left(k + c_0\left(d\log m + \log\left(\frac{1}{\delta}\right)\right)\right)}{m}\right)^{1/(d-1)}$$

$$\le \quad 8\left(\frac{\left(k + c_0\left(d\log m + \log\left(\frac{1}{\delta}\right)\right)\right)}{m}\right)^{1/(d-1)}$$

where, for the last inequality, we have used the fact that $d^{1/(2(d-1))} \le \sqrt{2}$ and for $d \ge 3$ it holds that $6^{1/(d-1)} \le \sqrt{6}$. $\qquad\square$

**Lemma D.1** ([Chaudhuri and Dasgupta, 2010]). *There is an absolute constant $c_0 > 0$ for which the following holds. Pick any $0 < \delta < 1$. Pick $\theta_1, \ldots, \theta_m$ independently at random from a distribution $\mu$ on $\mathbb{R}^d$. Then with probability at least $1 - \delta$, any ball $B$ in $\mathbb{R}^d$ with*

$$\nu(B) \ge \frac{2}{m}\left(k + c_0\left(d\log m + \log\left(\frac{1}{\delta}\right)\right)\right)$$

*contains at least $k$ of the $\theta_i$.*

**Lemma D.2** (Lemma 12 [Dasgupta and Tosh, 2020]). *Suppose $d \ge 3$, $r \in (0,1)$, and $\nu$ is the uniform distribution over $\mathcal{S}^{d-1}$. Then for any $x \in \mathcal{S}^{d-1}$,*

$$\nu(B(x,r)) \ge \frac{1}{3\sqrt{d}}r^{d-1}\left(1 - r^2/4\right)^{(d-1)/2} \ge \frac{1}{3\sqrt{d}}(3/4)^{(d-1)/2}r^{d-1}.$$

### D.3 Proof of Theorem 3.3

*Proof.* Since the weights $w_{n,i}$ given in equation 4 are non-negative, in order to satisfy condition (i) of Stone's theorem, we need to show that there exists a positive constant $c$ such that for any non-negative measurable function $f$ satisfying $\mathbb{E}f(x) < \infty$ and for any $n$, $\mathbb{E}\left(\sum_{i=1}^{n} w_{n,i}(x)f(x_i)\right) \le c\mathbb{E}(f(x))$. Using Lemma D.3, we show that this condition is satisfied for $c = 1$.

Concerning condition (ii) of Stone's theorem, first define $a_n = a_n(k_n, m_n, \delta_n) = 8\left(\frac{k_n + c_0\left(d\log m_n + \log\left(\frac{1}{\delta_n}\right)\right)}{m_n}\right)^{1/(d-1)}$, where $c_0$ is an absolute constant. Then $\lim_{n\to\infty}\delta_n = 0$ and using condition (i) and (ii) of this theorem we also have $\lim_{n\to\infty}a_n = 0$. Now pick any $a > 0$. We can always find a positive integer $N$ such that for $n > N$, we have $a > a_n$ satisfying $\sum_{i=1}^{n} w_{i,n}(x)\mathbb{I}_{\{\|x_i - x\| > a\}} \le \sum_{i=1}^{n} w_{i,n}(x)\mathbb{I}_{\{\|x_i - x\| > a_n\}}$ for all $x \in \mathcal{X}$. Replacing $k, m, \delta$ and $a$ with $k_n, \delta_n$ and $a_n$ respectively in Lemma D.6, we have

$$\mathbb{E}\left(\sum_{i=1}^{n} w_{i,n}(x)\mathbb{I}[\|x_i - x\| > a]\right) \le \mathbb{E}\left(\sum_{i=1}^{n} w_{i,n}(x)\mathbb{I}[\|x_i - x\| > a_n]\right) \le \delta_n$$

Taking limit yields, $\lim_{n\to\infty}\mathbb{E}\left(\sum_{i=1}^{n} w_{i,n}(x)\mathbb{I}[\|x_i - x\| > a]\right) = 0$.

Concerning condition (iii) of Stone's Theorem, note that

$$\mathbb{E}\left(\max_{1\le i\le n} w_{n,i}(x)\right) \quad \le \quad \mathbb{E}\left(\frac{1}{k}\sum_{j:x\in C_j}\frac{1}{\sum_{l=1}^{n}\mathbb{I}[x_l \in C_j]}\right) = \mathbb{E}\left(\frac{1}{k}\sum_{i=1}^{k}\frac{1}{\sum_{l=1}^{n}\mathbb{I}[x_i \in C_{j_i(x,\Theta)}]}\right)$$

$$= \quad \mathbb{E}\left(\frac{1}{k}\sum_{i=1}^{k}\frac{1}{N_{j_i(x,\Theta)}}\right) \le \mathbb{E}\left(\frac{1}{\min_{1\le i\le k}\{N_{j_i(x,\Theta)}\}}\right) \qquad (19)$$

where, for $i = 1, \ldots, k$, we have used the notation $j_i(x, \Theta)$ to denote the $k$ random coordinates of $h_{\Theta,k}(x)$ that are set to 1 and $N_{j_i(x,\Theta)}$ to denote the number of points from $x_1, \ldots, x_n$ that fall in $C_{j_i(x,\Theta)}$. Now from equation 19, is it easy to see that $\lim_{n \to \infty} \mathbb{E}\left(\max_{1 \le i \le n} w_{n,i}(x)\right) = 0$ since using lemma D.7, $\min_{1 \le i \le k}\{N_{j_i(x,\Theta)}\} \to \infty$ in probability, as $n \to \infty$. $\qquad \square$

## D.4 Technical results for satisfying condition (i) of Stone's theorem (Theorem 3.2)

**Lemma D.3.** *For any non-negative measurable function $f : \mathcal{X} \to \mathbb{R}_+$ satisfying $\mathbb{E} f(x) < \infty$ and for any $n$, $\mathbb{E}\left(\sum_{i=1}^n w_{n,i}(x) f(x_1)\right) \le \mathbb{E}(f(x))$, where the weights $w_{n,i}$ are as given in (4).*

*Proof.*

$$
\begin{aligned}
\mathbb{E}\left(\sum_{i=1}^n w_{i,n}(x) f(x_i)\right) &= \mathbb{E}\left(\sum_{i=1}^n \left(\frac{1}{k} \sum_{j:x \in C_j} \frac{\mathbb{I}[x_i \in C_j] f(x_i)}{\sum_{l=1}^n \mathbb{I}[x_l \in C_j]}\right)\right) \\
&= \mathbb{E}\left(\frac{1}{k} \sum_{j:x \in C_j} \left(\sum_{i=1}^n \frac{\mathbb{I}[x_i \in C_j] f(x_i)}{\sum_{l=1}^n \mathbb{I}[x_l \in C_j]}\right)\right) \\
&= \mathbb{E}_x\left(\frac{1}{k} \mathbb{E}\left[\sum_{j:x \in C_j} \left(\sum_{i=1}^n \frac{\mathbb{I}[x_i \in C_j] f(x_i)}{\sum_{l=1}^n \mathbb{I}[x_l \in C_j]}\right) \Big| x\right]\right) \\
&= \mathbb{E}_x\left(\frac{1}{k} \left[\sum_{j:x \in C_j} \mathbb{E}\left(\sum_{i=1}^n \frac{\mathbb{I}[x_i \in C_j] f(x_i)}{\sum_{l=1}^n \mathbb{I}[x_l \in C_j]} \Big| x\right)\right]\right) \\
&\overset{a}{\le} \mathbb{E}_x\left(\frac{1}{k}\left[\sum_{j:x \in C_j} \frac{1}{\mu(C_j)} \int_{C_j} f(x_1) \mu(dx_1)\right]\right) \\
&= \sum_{i=1}^{\binom{m}{k}}\left(\Pr(x \in C_i^k) \left[\frac{1}{k} \sum_{j \in \sigma(i)} \frac{1}{\mu(C_j)} \int_{C_j} f(x_1) \mu(dx_1)\right]\right) \\
&= \sum_{i=1}^{\binom{m}{k}} \frac{1}{k}\left[\sum_{j \in \sigma(i)} \frac{\Pr(x \in C_i^k)}{\mu(C_j)} \int_{C_j} f(x_1) \mu(dx_1)\right] \\
&= \frac{1}{k} \sum_{i=1}^{\binom{m}{k}} \sum_{j \in \sigma(i)} \frac{\mu(C_i^k)}{\mu(C_j)} \int_{C_j} f(x_1) \mu(dx_1) \\
&\overset{b}{=} \frac{1}{k} \sum_{j=1}^m \sum_{i:j \in \sigma(i)} \frac{\mu(C_i^k)}{\mu(C_j)} \int_{C_j} f(x_1) \mu(dx_1) \overset{c}{=} \frac{1}{k} \sum_{j=1}^m \int_{C_j} f(x_1) \mu(dx_1) \\
&\overset{d}{=} \frac{1}{k} \sum_{j=1}^m \sum_{i:j \in \sigma(i)} \int_{C_i^k} f(x_1) \mu(dx_1) \overset{e}{=} \frac{1}{k} \sum_i^{\binom{m}{k}} \sum_{j \in \sigma(i)} \left(\int_{C_i^k} f(x_1) \mu(dx_1)\right) \\
&= \frac{1}{k} \sum_i^{\binom{m}{k}} k \int_{C_i^k} f(x_1) \mu(dx_1) = \sum_i^{\binom{m}{k}} \int_{C_i^k} f(x_1) \mu(dx_1) \\
&\overset{f}{=} \int_{\S} f(x_1) \mu(dx_1) = \mathbb{E}(f(x_1)) \overset{g}{=} \mathbb{E}(f(x))
\end{aligned}
$$

where, inequality $a$ follows from Lemma D.4. Equality $b$ follows from the following observation. In the line above inequality $b$, we are summing $k \times \binom{m}{k}$ terms. Since $k \times \binom{m}{k} = m \times \binom{m-1}{k-1}$ and any $j \in [m]$ can appear in exactly $\binom{m-1}{k-1}$ different subsets of of size $k$, equality $b$ is simply rearranging the terms from the line above by changing the indices appropriately. Equality $c$ follows from part (iv)

of Lemma 3.4. Equality $d$ follows from part (iii) of Lemma 3.4. Equality $e$ follows from the following observation. In the line above equality $e$, we are summing $m \times \binom{m-1}{k-1}$ terms since any $j \in [m]$ can appear in exactly $\binom{m-1}{k-1}$ different subsets of of size $k$. Again noticing that $m \times \binom{m-1}{k-1} = k \times \binom{m}{k}$ and each $\sigma(i)$ has $k$ terms in it, we are simply rearranging the terms from the line above by changing the indices appropriately. Finally, equality $f$ follows from part (ii) of Lemma E.4 and equality $g$ follows from the fact that $x$ and $x_1$ are i.i.d. $\square$

**Lemma D.4.** $\mathbb{E}\left(\sum_{i=1}^{n} \frac{\mathbb{I}[x_i \in C_j]f(x_i)}{\sum_{l=1}^{n} \mathbb{I}[x_l \in C_j]}\Big|x\right) \leq \frac{1}{\mu(C_j)} \int_{C_j} f(x_1)\mu(dx_1)$.

*Proof.*

$$
\begin{aligned}
\mathbb{E}\left(\sum_{i=1}^{n} \frac{\mathbb{I}[x_i \in C_j]f(x_i)}{\sum_{l=1}^{n} \mathbb{I}[x_l \in C_j]}\Big|x\right) &= \sum_{i=1}^{n} \mathbb{E}\left(\frac{\mathbb{I}[x_i \in C_j]f(x_i)}{\sum_{l=1}^{n} \mathbb{I}[x_l \in C_j]}\Big|x\right) \\
&\stackrel{a}{=} \sum_{i=1}^{n} \mathbb{E}\left(\frac{\mathbb{I}[x_i \in C_j]f(x_i)}{1 + \sum_{l \neq i} \mathbb{I}[x_l \in C_j]}\Big|x\right) \\
&= n\mathbb{E}\left(I_{\{x_1 \in C_j\}}f(x_1)\frac{1}{1 + \sum_{l=2}^{n} \mathbb{I}[x_l \in C_j]}\Big|x\right) \\
&= n\mathbb{E}_{x_1}\left(I_{\{x_1 \in C_j\}}f(x_1)\mathbb{E}_{x_2,\ldots,x_n}\left[\frac{1}{1 + \sum_{l=2}^{n} \mathbb{I}[x_l \in C_j]}\Big|x, x_1\right]\right) \\
&\stackrel{b}{\leq} n\mathbb{E}_{x_1}\left(\mathbb{I}_{\{x_1 \in C_j\}}f(x_1)\frac{1}{n\mu(C_j)}\right) \\
&= \frac{1}{\mu(C_j)} \int_{C_j} f(x_1)\mu(dx_1)
\end{aligned}
$$

where, equality $a$ follows from that fact, that if $x_i \in C_j$, then the term in the parenthesis does not change and if $x_i \notin C_j$, then the term within the parenthesis is zero, and inequality $b$ follows from lemma D.5 since conditioned on $x$ and $x_1$, the random variable $\sum_{l=2}^{n} \mathbb{I}_{\{x_l \in \mathcal{C}_j\}}$ is Binomially distributed with parameters $(n-1)$ and $\mu(C_j)$. $\square$

**Lemma D.5.** *(Binomial bound Györfi et al. [2002]) Let the random variable $B(n,p)$ be Binomially distributed with parameters $n$ and $p$. Then,*

$$
\mathbb{E}\left(\frac{1}{1 + B(n,p)}\right) \leq \frac{1}{(n+1)p}
$$

## D.5 Technical results for satisfying condition (ii) of Stone's theorem (Theorem 3.2)

**Lemma D.6.** *Pick any $0 < \delta < 1$ and let $a = 8\left(\frac{k + c_0\left(d\log m + \log\left(\frac{1}{\delta}\right)\right)}{m}\right)^{1/(d-1)}$, where $c_0 > 0$ is an absolute constant defined in Lemma 3.5. Then,*

$$
\mathbb{E}\left(\sum_{i=1}^{n} w_{i,n}(x)\mathbb{I}[\|x_i - x\| > a]\right) \leq \delta.
$$

*Proof.* From Lemma 3.5, we have that with probability at least $1-\delta$, $\text{diam}(\mathcal{C}_j) \leq a$ for all $1 \leq j \leq m$. Let us define the random variable:

$$
\mathcal{A}_1 = \left\{\sum_{i=1}^{n}\left(\frac{1}{k}\sum_{j:x \in C_j} \frac{\mathbb{I}[x_i \in C_j]}{\sum_{i=1}^{n} \mathbb{I}[x_i \in C_j]}\right)\mathbb{I}[\|x_i - x\| > a]\right\}
$$

and the event:

$$
\mathcal{A}_2 = \left\{\cup_{j:x \in C_j} C_j \subset B(x,a)\right\}
$$

Let $\mathcal{A}_2^c$ denotes complement of the event $\mathcal{A}_2$. Then,

$$
\mathbb{E}\left(\sum_{i=1}^n w_{i,n}(x)\mathbb{I}[\|x_i - x\| > a]\right) \;=\; \mathbb{E}\left[\sum_{i=1}^n \left(\frac{1}{k}\sum_{j:x\in C_j}\frac{\mathbb{I}[x_i \in C_j]}{\sum_{i=1}^n \mathbb{I}[x_i \in C_j]}\right)\mathbb{I}_{\{\|x_i - x\|>a\}}\right]
$$

$$
=\; \mathbb{E}\left(\mathcal{A}_1|\mathcal{A}_2\right)\Pr\left(\mathcal{A}_2\right) + \mathbb{E}\left(\mathcal{A}_1|\mathcal{A}_2^c\right)\Pr\left(\mathcal{A}_2^c\right)
$$

$$
\overset{a}{\leq}\; 0\cdot\Pr(\mathcal{A}_2) + 1\cdot\delta = \delta
$$

where the inequality follows from the fact that maximum value of $\mathcal{A}_1$ is 1 and conditioned on the event $\mathcal{A}_2$, value of $\mathcal{A}_1$ is 0. $\qquad\square$

## D.6 Technical results for satisfying condition (iii) of Stone's theorem (Theorem 3.2)

**Lemma D.7.** *For any $x \sim \mu$, let $j_1(x,\Theta),\dots,j_k(x,\Theta) \in [m]$ be the $k$ random coordinates of $h_{\Theta,k}(x)$ that are set to 1. Let $N_{j_1(x,\Theta)},\dots,N_{j_k(x,\Theta)}$ be the number of data points falling in $C_{j_1(x,\Theta)},\dots,C_{j_k(x,\Theta)}$ respectively. Then $\min\{N_{j_1(x,\Theta)},\dots,N_{j_k(x,\Theta)}\} \to \infty$ in probability, whenever $n \to \infty$ and $m^k/n \to 0$ as $n \to \infty$.*

*Proof.* For any random $x \in \mathcal{X}$, let $C_{i(x,\Theta)}^k$ be the random cell of the partition $\{C_i^k\}_{i=1}^{\binom{m}{k}}$ which sets $k$ random coordinates $j_1(x,\Theta),\dots,j_k(x,\Theta)$ of $h_{\Theta,k}(x)$ to one. Then, $C_{i(x,\Theta)}^k = \cap_{i=1}^k C_{j_i(x,\Theta)}$. Let $N_n(x,\Theta) = \sum_{i=1}^n \mathbb{I}_{\{x_i \in C_{i(x,\Theta)}^k\}}$ be the number of data points falling in the same cell as $x$. We first we show that $N_n(x,\Theta) \to \infty$ in probability. Let $N_1, N_2, \dots N_{\binom{m}{k}}$ be the number of points of $x, x_1,\dots,x_n$ falling in the $\binom{m}{k}$ respective cells $C_1^k,\dots,C_{\binom{m}{k}}^k$. Let $S = \{x, x_1,\dots,x_n\}$ denote the set of positions of these $n+1$ points. Since these points are independent and identically distributed, fixing the set $S$ (but not the order of the points) and $\Theta$, the conditional probability that $x$ falls in the $i^{th}$ cell $C_i^k$ is $N_i/(n+1)$. Then for any fixed integer $t > 0$,

$$
\Pr(N_n(x,\Theta) < t) \;=\; \mathbb{E}\left[\Pr\left(N_n(x,\Theta) < t|S,\Theta\right)\right]
$$

$$
=\; \mathbb{E}\left[\sum_{i:N_i<t}\frac{N_i}{n+1}\right] \leq (t-1)\frac{\binom{m}{k}}{n+1} \leq (t-1)\frac{m^k}{n}
$$

which converges to zero on our assumption on $n$. Since $\mathcal{C}_{i(x,\Theta)}^k \subseteq \mathcal{C}_{j_l(x,\Theta)}$ for $l = 1,\dots,k$, we have $N_n(x,\Theta) \leq \min\{N_{j_1(x,\Theta)},\dots,N_{j_k(x,\Theta)}\}$ and the statement of the Lemma follows. $\qquad\square$

# E  Various proofs from section 3.2

For ease of exposition, we use the notation $x_{[1,n]}$ to denote $x_1, \ldots, x_n$ and $y_{[1,n]}$ to denote $y_1, \ldots, y_n$ for various proofs appearing in the section.

## E.1  Proof of lemma 3.8

*Proof.* Fix any $\Theta$. This ensures that $C_1, \ldots, C_m$ are fixed. Now,

$$\mathbb{E}(\hat{\eta}(x) - \bar{\eta}(x))^2$$
$$= \mathbb{E}_{x, x_{[1,n]}, y_{[1,n]}} \left[ (\hat{\eta}(x) - \bar{\eta}(x))^2 \right]$$
$$= \mathbb{E}_{x_{[1,n]}} \left[ \mathbb{E}_{x, y_{[1,n]}} \left[ (\hat{\eta}(x) - \bar{\eta}(x))^2 | x_{[1,n]} ) \right] \right]$$
$$= \mathbb{E}_{x_{[1,n]}} \left[ \mathbb{E}_{x, y_{[1,n]}} \left[ \left( \frac{1}{k} \sum_{j:x \in C_j} (\hat{\eta}_j - \bar{\eta}_j) \right)^2 \Big| x_{[1,n]} \right] \right]$$
$$\overset{a}{\leq} \mathbb{E}_{x_{[1,n]}} \left[ \mathbb{E}_{x, y_{[1,n]}} \left[ \frac{1}{k} \sum_{j:x \in C_j} (\hat{\eta}_j - \bar{\eta}_j)^2 \Big| x_{[1,n]} \right] \right]$$
$$= \mathbb{E}_{x_{[1,n]}} \left[ \mathbb{E}_{x, y_{[1,n]}} \left[ \frac{1}{k} \sum_{j:x \in C_j} \left( \frac{\sum_{i=1}^{n} Y_i \mathbb{I}[x_i \in C_j]}{\sum_{i=1}^{n} \mathbb{I}[x_i \in C_j]} - \eta(C_j) \right)^2 \Big| x_{[1,n]} \right] \right]$$
$$= \mathbb{E}_{x_{[1,n]}} \left[ \mathbb{E}_{x, y_{[1,n]}} \left[ \frac{1}{k} \sum_{j:x \in C_j} \left( \frac{\sum_{i=1}^{n} y_i \mathbb{I}[x_i \in C_j]}{n\mu_n(C_j)} - \eta(C_j) \right)^2 \Big| x_{[1,n]} \right] \right]$$
$$= \mathbb{E}_{x_{[1,n]}} \left[ \mathbb{E}_{x, y_{[1,n]}} \left[ \frac{1}{k} \sum_{j:x \in C_j} \left( \frac{\sum_{i=1}^{n} y_i \mathbb{I}[x_i \in C_j]}{n\mu_n(C_j)} - \eta(C_j) \right)^2 \mathbb{I}[n\mu_n(C_j) > 0] + \eta^2(C_j)\mathbb{I}[\mu_n(C_j) = 0] \Big| x_{[1,n]} \right] \right]$$
$$= \mathbb{E}_{x_{[1,n]}} \left[ \mathbb{E}_{x, y_{[1,n]}} \left[ \frac{1}{k} \sum_{j:x \in C_j} \left( \frac{\sum_{i=1}^{n} (y_i - \eta(C_j))\mathbb{I}[x_i \in C_j]}{n\mu_n(C_j)} \right)^2 \mathbb{I}[n\mu_n(C_j) > 0] \Big| x_{[1,n]} \right] \right]$$
$$\qquad\qquad + \mathbb{E}_{x_{[1,n]}} \left[ \mathbb{E}_{x, y_{[1,n]}} \left[ \frac{1}{k} \sum_{j:x \in C_j} \eta^2(C_j)\mathbb{I}[\mu_n(C_j) = 0] \Big| x_{[1,n]} \right] \right]$$
$$\overset{b}{\leq} \mathbb{E}_{x_{[1,n]}} \left[ \mathbb{E}_{x, y_{[1,n]}} \left[ \frac{1}{k} \sum_{h_{\Theta,k}(x)[j]=1} \left( \frac{\sum_{i=1}^{n} (y_i - \eta(C_j))\mathbb{I}[x_i \in C_j]}{n\mu_n(C_j)} \right)^2 \mathbb{I}[n\mu_n(C_j) > 0] \Big| x_{[1,n]} \right] \right] + \frac{m}{kne}$$
$$= \mathbb{E}_{x_{[1,n]}} \left[ \mathbb{E}_x \left[ \frac{1}{k}\mathbb{E}_{y_{[1,n]}} \left[ \sum_{j:x \in C_j} \left( \frac{\sum_{i=1}^{n} (y_i - \eta(C_j))\mathbb{I}[x_i \in C_j]}{n\mu_n(C_j)} \right)^2 \mathbb{I}[n\mu_n(C_j) > 0] \Big| x, x_{[1,n]} ) \right] \right] \right] + \frac{m}{kne}$$
$$\overset{c}{\leq} \frac{1}{4}\mathbb{E}_{x_{[1,n]}} \left[ \mathbb{E}_x \left( \frac{1}{k} \sum_{j:x \in C_j} \frac{\mathbb{I}[n\mu_n(C_j) > 0]}{n\mu_n(C_j)} \Big| X_{[1,n]} \right) \right] + \frac{m}{kne}$$
$$\overset{d}{\leq} \frac{1}{4}\mathbb{E}_{x_{[1,n]}} \left[ \frac{1}{k} \sum_{j=1}^{m} \left( \frac{\mu(C_j)\mathbb{I}[n\mu_n(C_j) > 0]}{n\mu_n(C_j)} \right) \right] + \frac{m}{kne}$$
$$= \frac{1}{4k} \sum_{j=1}^{m} \mu(C_j)\mathbb{E}_{x_{[1,n]}} \left( \frac{\mathbb{I}[n\mu_n(C_j) > 0]}{n\mu_n(C_j)} \right) + \frac{m}{kne}$$
$$\overset{e}{\leq} \frac{1}{4k} \sum_{j=1}^{m} \frac{2\mu(C_j)}{(n+1)\mu(C_j)} + \frac{m}{kne} = \frac{m}{2k(n+1)} + \frac{m}{kne} \leq \frac{m}{2kn} + \frac{m}{kne} \leq \frac{m}{kn}.$$

where, inequality $a$ is due to Jensen's inequality. Inequality $b$ follows from lemma E.3. Inequality $c$ follows from lemma E.1 and inequality $d$ follows from lemma E.2. Finally inequality $e$ follow from the observation that $n\mu_n(C_j)$ is Binomially distributed with parameters $n$ and $\mu(C_l)$ and by an application of lemma E.4. The result follows noting that by Jensen's inequality $\mathbb{E}|\hat{\eta}(x) - \bar{\eta}(x)| \leq \sqrt{\mathbb{E}(\hat{\eta}(x) - \bar{\eta}(x))^2}$. $\qquad\square$

**Lemma E.1.** *Pick $m \times d$ projection matrix $\Theta$. Suppose the EaS representation uses (i) a mapping $\Theta$ and (ii) k-winner-take-all sparsification. Let $x$ be sampled from $\mu$ and let $D_n = ((x_1, y_1), \ldots, (x_n, y_n))$ is a random training set where $x_i$ is sampled from $\mu$ and $y_i$ is distributed as $\eta(x_i)$ for $i \in [n]$. Then the following holds.*

$$\mathbb{E}_{y_{[1,n]}}\left[\sum_{j:x\in C_j}\left(\frac{\sum_{i=1}^n(y_i-\eta(C_j))\mathbb{I}[x_i\in C_j]}{n\mu_n(C_j)}\right)^2\mathbb{I}[n\mu_n(C_j)>0]\Big|x,x_{[1,n]}\right]\leq\frac{1}{4}\sum_{j:x\in C_j}\left(\frac{\mathbb{I}[n\mu_n(C_j)>0]}{n\mu_n(C_j)}\right)$$

*Proof.* Conditioned on $x$, only $k$ of the $m$ coordinates in the EaS representation of $x$ are non-zero. WLOG, for ease of exposition, assume these $k$ non-zero coordinate to be $j_1, \ldots, j_k \in [m]$. Then the number of $x_i$ that falls in any such $C_{j_l}$, where $l \in [k]$, is $n\mu_n(C_{j_l})$. The $y_i$ values corresponding to these $x_i$ points (there are $n\mu_n(C_{j_l})$ of them in total) are identically and independently distributed with expectation

$$\mathbb{E}(y_i|x_i\in C_{j_l}) \quad = \quad \Pr(y_i=1|x_i\in C_{j_l}) = \frac{1}{\mu(C_{j_l})}\int_{C_{j_l}}\Pr(y_i=1|x_i=x)\mu(dx)$$

$$= \quad \frac{1}{\mu(C_{j_l}}\int_{C_{j_l}}\eta(x)\mu(dx) = \eta(C_{j_l})$$

Therefore, we can write

$$\mathbb{E}_{y_{[1,n]}}\left[\sum_{j:x\in C_j}\left(\frac{\sum_{i=1}^n(y_i-\eta(C_j))\mathbb{I}[x_i\in C_j]}{n\mu_n(C_j)}\right)^2\mathbb{I}[n\mu_n(C_j)>0]\Big|x,x_{[1,n]}\right]$$

$$=\sum_{l=1}^k\left[\frac{\mathbb{E}_{y_{[1,n]}}\left[(\sum_{i=1}^n(y_i-\eta(C_{j_l}))\mathbb{I}[x_i\in C_{j_l}])^2\mathbb{I}[n\mu_n(C_{j_l})>0]|x_{[1,n]}\right]}{(n\mu_n(C_{j_l}))^2}\right]$$

$$\overset{a}{=}\sum_{l=1}^k\left[\frac{\sum_{i=1}^n\mathbb{E}_{y_i}\left[(y_i-\eta(C_{j_l}))^2\mathbb{I}[x_i\in C_{j_l}]\mathbb{I}[n\mu_n(C_{j_l})>0]|x_i\right]}{(n\mu_n(C_{j_l}))^2}\right]$$

$$\overset{b}{=}\sum_{l=1}^k\left[\frac{\sum_{i=1}^n\eta(C_{j_l})(1-\eta(C_{j_l}))\mathbb{I}[x_i\in C_{j_l}]\mathbb{I}[n\mu_n(C_{j_l})>0]}{(n\mu_n(C_{j_l}))^2}\right]$$

$$\overset{c}{\leq}\frac{1}{4}\sum_{l=1}^k\left(\frac{\mathbb{I}[n\mu_n(C_{j_l})>0]}{n\mu_n(C_{j_l})}\right)$$

where, equality $a$ is due to the following observation. For any $i, j \in [m], i \neq j$ and $x_i, x_j \in C_l$ for some $l \in [m]$, $y_i$ and $y_j$ are identically and independently distributed with expectation $\eta(C_l)$. Therefore, the expectation of the cross product is simply:

$$\mathbb{E}_{y_i,y_j}\left[(y_i-\eta(C_l))(y_j-\eta(C_l))\right] \quad = \quad \mathbb{E}_{y_i,y_j}\left[y_iy_j-\eta(C_l)(y_i+y_j)+\eta(C_l)^2\right]$$

$$= \quad \mathbb{E}y_i\mathbb{E}y_j-\eta(C_l)(\mathbb{E}y_i+\mathbb{E}y_j)+\eta(C_l)^2=0$$

Equality $b$ follows from variance computation. In particular for any $Y_i, i \in [m]$ with $x_i \in C_l$ for some $l \in [m]$, $\mathbb{E}\left[(y_i-\eta(C_l))^2\right] = \mathbb{E}y_i^2 - 2\eta(C_l)\mathbb{E}y_i + \eta(C_l)^2 = \eta(C_l) - 2\eta(C_l)^2 + \eta(C_l)^2 = \eta(C_l)(1-\eta(C_l))$. Finally, inequality $c$ follows from the fact that for any $z \in [0,1]$, the maximum value of $z(1-z)$ is $\frac{1}{4}$.

It is easy to observe that the final result is equivalent to $\frac{1}{4}\sum_{j:x\in C_j}\left(\frac{\mathbb{I}[n\mu_n(C_j)>0]}{n\mu_n(C_j)}\right)$. $\qquad\square$

**Lemma E.2.** *Pick $m \times d$ projection matrix $\Theta$. Suppose the EaS representation uses (i) a mapping $\Theta$ and (ii) k-winner-take-all sparsification. Let $x$ be sampled from $\mu$ and let $D_n = ((x_1, y_1), \ldots, (x_n, y_n))$ is a random training set where $x_i$ is sampled from $\mu$ and $y_i$ is distributed as $\eta(x_i)$ for $i \in [n]$. Then conditioned on $x_1, \ldots, x_n$, the following holds.*

$$\mathbb{E}_x \left( \frac{1}{k} \sum_{j:x \in C_j} \frac{\mathbb{I}_{\{n\mu_n(C_j)>0\}}}{n\mu_n(C_j)} \right) \leq \frac{1}{k} \sum_{j=1}^{m} \left( \frac{\mu(C_j)\mathbb{I}_{\{n\mu_n(C_j)>0\}}}{n\mu_n(C_j)} \right)$$

*Proof.*

$$
\begin{aligned}
\mathbb{E}_x \left( \frac{1}{k} \sum_{j:x \in C_j} \frac{\mathbb{I}[n\mu_n(C_j) > 0]}{n\mu_n(C_j)} \right) &\overset{a}{=} \sum_{i=1}^{\binom{m}{k}} \left( \Pr(x \in C_i^k) \left( \frac{1}{k} \sum_{j \in \sigma(i)} \frac{\mathbb{I}[n\mu_n(C_j) > 0]}{n\mu_n(C_j)} \right) \right) \\
&= \sum_{i=1}^{\binom{m}{k}} \left( \frac{1}{k} \left( \sum_{j \in \sigma(i)} \frac{\mu(C_i^k)\mathbb{I}[n\mu_n(C_j) > 0]}{n\mu_n(C_j)} \right) \right) \\
&\overset{b}{=} \frac{1}{k} \sum_{j=1}^{m} \sum_{i:j \in \sigma(i)} \left( \frac{\mu(C_i^k)\mathbb{I}[n\mu_n(C_j) > 0]}{n\mu_n(C_j)} \right) \\
&= \frac{1}{k} \sum_{j=1}^{m} \left( \frac{\mathbb{I}[n\mu_n(C_j) > 0]}{n\mu_n(C_j)} \right) \sum_{i:j \in \sigma(i)} \mu(C_i^k) \\
&\overset{c}{=} \frac{1}{k} \sum_{j=1}^{m} \left( \frac{\mu(C_j)\mathbb{I}[n\mu_n(C_j) > 0]}{n\mu_n(C_j)} \right)
\end{aligned}
$$

where equality $a$ follows from part (ii) of lemma 3.4 since $\{C_i^k\}_{i=1}^{\binom{m}{k}}$ forms a partition of $\mathcal{X}$ and the definition of $\sigma$ in section 3.2. Equality $b$ follows from the following observation. In the line above inequality $b$, we are summing $k \times \binom{m}{k}$ terms. Since $k \times \binom{m}{k} = m \times \binom{m-1}{k-1}$ and any $j \in [m]$ can appear in exactly $\binom{m-1}{k-1}$ different subsets of of size $k$, equality $b$ is simply rearranging the terms from the line above by changing the indices appropriately. Equality $c$ follows from part (iv) of lemma 3.4. $\qquad \square$

**Lemma E.3.** *Pick $m \times d$ projection matrix $\Theta$. Suppose the EaS representation uses (i) a mapping $\Theta$ and (ii) k-winner-take-all sparsification. Let $x$ be sampled from $\mu$ and let $D_n = ((x_1, y_1), \ldots, (x_n, y_n))$ is a random training set where $x_i$ is sampled from $\mu$ and $y_i$ is distributed as $\eta(x_i)$ for $i \in [n]$. Then the following holds.*

$$\mathbb{E}_{x_{[1,n]}} \left[ \mathbb{E}_{x,y_{[1,n]}} \left[ \frac{1}{k} \sum_{j:x \in C_j} \eta^2(C_j)\mathbb{I}[\mu_n(C_j) = 0] \Big| x_{[1,n]} \right] \right] \leq \frac{m}{nke}$$

*Proof.*

$$
\mathbb{E}_{x_{[1,n]}} \left[ \mathbb{E}_{x,y_{[1,n]}} \left[ \frac{1}{k} \sum_{j:X \in C_j} \eta^2(C_j) \mathbb{I}[\mu_n(C_j) = 0] \Big| x_{[1,n]} \right] \right]
$$

$$
\overset{a}{=} \mathbb{E}_{x_{[1,n]}} \left[ \mathbb{E}_x \left[ \frac{1}{k} \sum_{j:x \in C_j} \eta^2(C_j) \mathbb{I}[\mu_n(C_j) = 0] \Big| x_{[1,n]} \right] \right]
$$

$$
\overset{b}{=} \mathbb{E}_{x_{[1,n]}} \left[ \sum_{i=1}^{\binom{m}{k}} \mathrm{Pr}(x \in C_i^k) \left( \frac{1}{k} \sum_{j \in \sigma(i)} \eta^2(C_j) \mathbb{I}[\mu_n(C_j) = 0] \right) \right]
$$

$$
= \mathbb{E}_{x_{[1,n]}} \left[ \frac{1}{k} \sum_{i=1}^{\binom{m}{k}} \sum_{j \in \sigma(i)} \mu(C_i^k) \eta^2(C_j) \mathbb{I}[\mu_n(C_j) = 0] \right]
$$

$$
\overset{c}{=} \mathbb{E}_{x_{[1,n]}} \left[ \frac{1}{k} \sum_{j=1}^{m} \eta^2(C_j) \mathbb{I}[\mu_n(C_j) = 0] \left( \sum_{i:j \in \sigma(i)} \mu(C_i^k) \right) \right]
$$

$$
= \frac{1}{k} \sum_{j=1}^{m} \eta^2(C_j) \mu(C_j) \left[ \mathbb{E}_{x_1,\dots,x_n} \mathbb{I}[\mu_n(C_j) = 0] \right]
$$

$$
= \frac{1}{k} \sum_{j=1}^{m} \eta^2(C_j) \mu(C_j) (1 - \mu(C_j))^n = \frac{1}{nk} \sum_{j=1}^{m} \eta^2(C_j) n \mu(C_j) (1 - \mu(C_j))^n
$$

$$
\leq \frac{1}{nk} \sum_{j=1}^{m} \eta^2(C_j) n \mu(C_j) e^{-n\mu(C_j)}
$$

$$
\overset{d}{\leq} \frac{1}{nk} \sum_{j=1}^{m} n\mu(C_j) e^{-n\mu(C_j)} \leq \frac{m}{nk} \max_j \left\{ n\mu(C_j) e^{-n\mu(C_j)} \right\} \overset{e}{\leq} \frac{m}{nke}
$$

where equality $a$ follows from the fact that the quantity with the inner square bracket is unaffected by the $Y_i$s. Equality $b$ follows from part (ii) of lemma 3.4 since $\{C_i^k\}_{i=1}^{\binom{m}{k}}$ forms a partition of $\mathcal{X}$ and the definition of $\sigma$ in section 3.2. Equality $c$ follows from the following observation. In the line above inequality $c$, we are summing $k \times \binom{m}{k}$ terms. Since $k \times \binom{m}{k} = m \times \binom{m-1}{k-1}$ and any $j \in [m]$ can appear in exactly $\binom{m-1}{k-1}$ different subsets of of size $k$, equality $c$ is simply rearranging the terms from the line above by changing the indices appropriately. Inequality $d$ follows from the fact $\max_j \eta(C_j) \leq 1$. Finally inequality $e$ follows from the fact that $\sup_z ze^{-z} = \frac{1}{e}$. $\qquad\square$

**Lemma E.4.** *(Binomial bound Györfi et al. [2002]) Let the random variable $B(n,p)$ be Binomially distributed with parameters $n$ and $p$. Then,*

$$
\mathbb{E} \left( \frac{1}{B(n,p)} \mathbb{I}[B(n,p) > 0] \right) \leq \frac{2}{(n+1)p}
$$

## E.2 Proof of Lemma 3.10

*Proof.* Take any $x \in \mathcal{X}$. Then

$$
|\eta(x) - \bar{\eta}(x)| = \left| \eta(x) - \frac{1}{k} \sum_{j:x \in C_j} \bar{\eta}_j \right| = \left| \frac{1}{k} \sum_{x \in C_j} (\eta(x) - \eta(x')) \right| \leq \frac{1}{k} \sum_{j:x \in C_j} |\eta(x) - \bar{\eta}_j|
$$

$$
= \frac{1}{k} \sum_{j:x \in C_j} \left| \eta(x) - \frac{1}{\mu(C_j)} \int_{C_j} \eta(x')\mu(dx') \right|
$$

$$
= \frac{1}{k} \sum_{j:x \in C_j} \left| \frac{1}{\mu(C_j} \int_{C_j} (\eta(x) - \eta(x'))\mu(dx') \right|
$$

$$
\leq \frac{1}{k} \sum_{j:x \in C_j} \frac{1}{\mu(C_j)} \int_{C_j} |\eta(x) - \eta(x')|\mu(dx')
$$

$$
\leq \frac{1}{k} \sum_{j:x \in C_j} \frac{1}{\mu(C_j)} \int_{C_j} L\|x - x'\|^\beta \mu(dx')
$$

$$
\leq \frac{1}{k} \sum_{j:x \in C_j} \frac{L \cdot (\mathrm{diam}(C_j))^\beta}{\mu(C_j)} \int_{C_j} \mu(dx') \leq L \cdot \max_{j \in [m]} (\mathrm{diam}(C_j))^\beta
$$

$\square$

## E.3 Proof of Theorem 3.15

*Proof.* Define $\eta : \mathcal{X}_1 \to [0,1]$ to be a triangular function defined below: for $0 < \theta \leq 2\pi$,

$$
\eta(\cos\theta, \sin\theta, 0, 0, \ldots, 0) = \begin{cases} \frac{\theta}{\pi}, & \text{if } \theta \leq \pi \\ 2 - \frac{\theta}{\pi}, & \text{if } \theta > \pi \end{cases}
$$

Clearly, $\eta$ is $\frac{1}{2}$-Lipschitz and for $k = 1$, $\bar{\eta}(x) = \sum_{j=1}^m \eta(C_j)\mathbb{I}_{\{x \in C_j\}}$. Therefore, using Theorem E.5, with probability at least $1/2$ over the choice of $\Theta$,

$$
\sup_{x \in \mathcal{X}_1} |\eta(x) - \bar{\eta}(x)| \geq c_d'' \cdot \frac{1}{m^{1/(d-1)}\log m}
$$

where $c_d''$ is am absolute constant depending on $d$. Taking expectation, with probability at least $1/2$ over the choice of $\Theta$, we have, $\mathbb{E}_{X,D_n}|\bar{\eta}(X) - \eta(X)| \geq c_d'' \cdot \frac{1}{m^{1/(d-1)}\log m}$. Next, using Lemma 3.8 with $k = 1$ yields, $\mathbb{E}_{X,D_n}|\hat{\eta}(X) - \bar{\eta}(X)| \leq \sqrt{\frac{m}{n}}$.

Using triangle inequality combining these results,

$$
\begin{aligned}
\mathbb{E}_{X,D_n}|\eta(X) - \hat{\eta}(X)| &= \mathbb{E}_{X,D_n}|\eta(X) - \bar{\eta}(X) + \bar{\eta}(X) - \hat{\eta}(X)| \\
&\geq \mathbb{E}_{X,D_n}|\bar{\eta}(X) - \eta(X)| - \mathbb{E}_{X,D_n}|\hat{\eta}(X) - \bar{\eta}(X)| \\
&\geq c_d'' \cdot \frac{1}{m^{1/(d-1)}\log m} - \sqrt{\frac{m}{n}} \qquad (20)
\end{aligned}
$$

Ignoring the log term we see that for large enough $n$, the first term on the right hand side of (20) will dominate. In particular, for $n = \Omega\left(m^{1+\frac{2}{d-1}}\right)$, we get $\mathbb{E}_{X,D_n}|\eta(X) - \hat{\eta}(X)| \geq \Omega\left(m^{-\frac{1}{d-1}}\right)$. To see this (ignoring the log term), if $\sqrt{\frac{m}{n}} \leq \frac{c_d''}{2} \cdot \frac{1}{m^{1/(d-1)}}$, then $\mathbb{E}_{X,D_n}|\eta(X) - \hat{\eta}(X)| \geq \frac{c_d''}{2} \cdot \frac{1}{m^{1/(d-1)}}$. Now,

$$
\sqrt{\frac{m}{n}} \leq \frac{c_d''}{2} \cdot \frac{1}{m^{1/(d-1)}} \Rightarrow n \geq \frac{4}{(c_d'')^2} \cdot m^{1+\frac{2}{d-1}}.
$$

In particular, setting $n = \frac{4}{(c_d'')^2} m^{1 + \frac{2}{d-1}}$, we have

$$\mathbb{E}_{X, D_n} |\eta(X) - \hat{\eta}(X)| \geq \frac{c_d''}{2} m^{-\frac{1}{d-1}} = \left( \frac{c_d''}{2} \right)^{\frac{d-3}{d-1}} n^{-\frac{1}{d+1}}.$$

$\square$

**Theorem E.5.** *[Theorem 4 of Dasgupta and Tosh [2020]] For any $d > 3$, let input space $\mathcal{X}_1$ be the one-dimensional sub-manifold of $\mathbb{R}^d$ given in (12). Take $k = 1$. Suppose that random matrix $\Theta$ has rows chosen from the distribution $\nu$ that is uniform over $\mathcal{S}^{d-1}$. For any $0 < \lambda < 1$, there exists a $\lambda$-Lipschitz function $f : \mathcal{X}_1 \to \mathbb{R}$ such that with probability at least $1/2$ over the choice of $\Theta$, no matter how the weights $w_1, \ldots, w_m$ are set, the resulting function $\hat{f}(x) = \sum_{j=1}^m w_j \mathbb{I}_{\{x \in C_j\}}$ has approximation error at least*

$$\sup_{x \in \mathcal{X}_1} |\hat{f}(x) - f(x)| \geq c_d' \cdot \lambda \cdot \frac{1}{m^{1/(d-1)} \log m}$$

*where $c_d'$ is some absolute constant depending on $d$.*

# F    Various proofs from section 4 and Appendix B

## F.1    Lemma F.1 and its proof

We first show that while EaS representation using $h_2$ is not $k$-sparse, for large enough sample size, with high probability, it is at least $k/2$-sparse and at most $2k$-sparse in expectation. F).

**Lemma F.1.** *Pick any $\delta > 0$. For $k, m$ and $n$, satisfying $n \geq \frac{c_0 m}{k} \left( \log n + \log \left( \frac{m}{\delta} \right) \right)$, where $c_0 > 0$ is a universal constant, with probability at least $1 - \delta$, over the random choice of training set $D_n'$, the following holds for all $j \in [m]$.*

$$(k/2m) \leq \Pr_{x \sim \mu}(\theta_j \cdot x \geq \tau_n(\theta_j)) \leq (2k/m).$$

*Proof.* We start with a version of relative generalization error bound, originally due to Vapnik and Chervonenkis, that appeared in Chaudhuri and Dasgupta [2010].

**Theorem F.2.** *(Theorem 15 of Chaudhuri and Dasgupta [2010]) Let $\mathcal{G}$ be a class of functions from $\mathcal{X}$ to $\{0, 1\}$ with VC dimension $d < \infty$, and $\mathbb{P}$ a probability distribution on $\mathcal{X}$. Let $\mathbb{E}$ be the expectation with respect to $\mathbb{P}$. Suppose $n$ points are drawn independently from $\mathbb{P}$; let $\mathbb{E}_n$ denotes expectation with respect to this sample. Then for any $\delta > 0$, with probability at least $1 - \delta$ the following holds for all $g \in \mathcal{G}$:*

$$-\min \left( \beta_n \sqrt{\mathbb{E}_n g}, \beta_n^2 + \beta_n + \sqrt{\mathbb{E}g} \right) \leq \mathbb{E}g - \mathbb{E}_n g \leq \min \left( \beta_n^2 + \beta_n \sqrt{\mathbb{E}_n g}, \beta_n \sqrt{\mathbb{E}g} \right)$$

*where $\beta_n = \sqrt{\frac{4(d \log(2n) + \log(8/\delta))}{n}}$.*

Let $\theta_i$ be the $i^{th}$ row of the projection matrix $\Theta$ and let $\mu^i$ be the distribution of $\theta_i \cdot x$ where $x$ is distribution according to $\mu$. Then, for any $a \in \mathbb{R}$,

$$\Pr_{x \sim \mu}(\theta_i \cdot x \geq a) = \mu^i(\theta_i \cdot x \geq a).$$

Suppose $x_1', \ldots, x_n'$ are sampled independently at random from $\mu$. For any interval $A_a = [a, \infty)$, let

$$\mu_n^i(A_a) = \frac{1}{n} \sum_{j=1}^n \mathbb{I}[\theta_i \cdot x_j' \geq a].$$

Consider the class of indicator functions over the intervals $[a, \infty)$ for $a \in \mathbb{R}$. This class has VC dimension 1 and define $\beta^2 = \frac{4}{n} \left( \log 2n + \log(\frac{8m}{\delta}) \right)$. Suppose $\mu_n^i(A_a)$ satisfies the condition $\mu_n^i(A_a) \geq 4\beta^2$.

The bound $\mu^i(A_a) - \mu_n^i(A_a) \leq \beta^2 + \beta\sqrt{\mu_n^i(A_a)}$ from Theorem F.2 yields,

$$\mu^i(A_a) \leq \mu_n^i(A_a) + \beta^2 + \beta\sqrt{\mu_n^i(A_a)} \leq \mu_n^i(A_a) + \frac{1}{4} \cdot \mu_n^i(A_a) + \frac{1}{2} \cdot \mu_n^i(A_a) = \frac{7}{4} \cdot \mu_n^i(A_a)$$

Similarly, the bound $-\beta\sqrt{\mu_n^i(A_a)} \leq \mu^i(A_a) - \mu_n^i(A_a)$ from Theorem F.2 yields,

$$\mu_i(A_a) \geq \mu_n^i(A_a) - \beta\sqrt{\mu_n^i(A_a)} \leq \mu_n^i(A_a) - \frac{1}{2} \cdot \mu_a^i(A_a) = \frac{1}{2} \cdot \mu_n^i(A_a)$$

Combining these two bounds, we can can conclude that for all intervals $A_a$ satisfying $\mu_n^i(A_a) \geq 4\beta^2$, with probability at least $1 - \frac{\delta}{m}$ it holds that

$$\frac{1}{2} \cdot \mu_n^i(A_a) \leq \mu^i(A_a) \leq \frac{7}{4} \cdot \mu_n^i(A_a).$$

Now if we only consider the intervals $A_{\tau_n(\theta_j)}$, then from (13) it is clear that $\mu_n^i(A_{\tau_n(\theta_j)}) \leq \frac{k}{m} + \frac{1}{n}$. This is because for any interval $\mu_n^i(\cdot)$ can take $(n+1)$ possible values in steps of $\frac{1}{n}$, namely, $0, \frac{1}{n}, \frac{2}{n}, \ldots, 1$. Therefore, from the definition of $\tau_n(\theta_j)$ in (13), if $\mu_n^i(A_{\tau_n(\theta_j)}) \neq \frac{k}{m}$, then to ensure that $\mu_n^i(A_{\tau_n(\theta_j)}) \geq \frac{k}{m}$, its maximum value can be at most $\frac{k}{m} + \frac{1}{n}$. Therefore, the condition $\mu_n^i(A_{\tau_n(\theta_j)}) \geq 4\beta^2$ implies,

$$\frac{k}{m} + \frac{1}{n} \geq \frac{16}{n}\left(\log 2n + \log\left(\frac{8m}{\delta}\right)\right) = \frac{16}{n}\left(\log 2 + \log n + \log 8 + \log\left(\frac{m}{\delta}\right)\right)$$

$$= \frac{64}{n} + \frac{16}{n}\left(\log n + \log\left(\frac{m}{\delta}\right)\right)$$

or in other words, $\frac{k}{m} \geq \frac{63}{n} + \frac{16}{n}\left(\log n + \log\left(\frac{m}{\delta}\right)\right)$. Noting that,

$$\frac{7}{4} \cdot \mu_n^i(A_{\tau_n(\theta_j)}) \leq \frac{7k}{4m} + \frac{7}{4n} \leq \frac{7k}{4m} + \frac{7}{4} \cdot \frac{k}{63m} \leq \frac{2k}{m}.$$

Therefore, if $\frac{k}{m} \geq \frac{c_0}{n}\left(\log n + \log\left(\frac{m}{\delta}\right)\right)$ for some universal constant $c_0 > 0$, then with probability at least $1 - \frac{\delta}{m}$,

$$\frac{k}{2m} \leq \Pr_{x\sim\mu}\left(\theta_j \cdot x \geq \tau_n(\theta_j)\right) \leq \frac{2k}{m}.$$

Taking union bound over $\theta_1, \ldots, \theta_m$ yields the desired results. $\qquad\square$

## F.2 Proof of Lemma B.3

*Proof.* Pick any good $\theta$, and let $x = \pi_M(\theta)$ be its projection on $M$. Since $\mathcal{X}$ consists of unit vectors, using Lemma F.1 with probability at least $1 - \delta$, the points that lie in $C(\theta)$ are at least $k/(2m)$ fraction or at most $2k/m$ fraction of $x$'s (under distribution $\mu$) that have highest dot product with $\theta$ or equivalent to closest to $\theta$. Thus, $C(\theta)$ is the set of the form $B(\theta, r')$ where radius $r'$ is so chosen that $k/(2m) \leq \mu(B(\theta, r')) \leq 2k/m$. However, it is not of the form $B(x, r'')$ which causes complication. In particular, two questions need to be answered: (i) if a point $x' \in M$ lies within distance $r < \rho$ of $x$, how far can it possibly be from $\theta$, and conversely, (ii) if $x' \in M$ lies within distance $r < \rho$ of $\theta$, how far can it possibly be from $x$? These two questions are answered in Lemma 6 of Dasgupta and Tosh [2020] showing that,

$$B_M(x, r) \subset B\left(\theta, \sqrt{\Delta^2 + \frac{\rho+\Delta}{\rho}r^2}\right) \tag{21}$$

$$B_M(\theta, r') \subset B\left(x, \sqrt{\frac{\rho}{\rho-\Delta}((r')^2 - \Delta^2)}\right) \tag{22}$$

For the left-hand containment pick $r = \sqrt{\frac{\rho-\Delta}{\rho+\Delta}}\left(\frac{k}{2c_2 m}\right)^{1/d_0}$. Further taking,

$$r' = \sqrt{\Delta^2 + \frac{\rho+\Delta}{\rho}r^2}$$

$$r'' = \sqrt{\frac{\rho}{\rho - \Delta}((r')^2 - \Delta^2)} = \left(\frac{k}{2c_2 m}\right)^{1/d_0}$$

and using (17) yields that $\mu(B_M(x, r'')) < k/(2m)$. Now using (21) and (22) we have, $B_M(x, r) \subset B_M(\theta, r') \subset B_M(x, r'')$. Since $\mu(B_M(x, r'')) < k/(2m)$, this ensures $B_M(x, r) \subset B_M(\theta, r') \subset C(\theta)$.

Pick $r = \left(\frac{2k}{c_1 m}\right)^{1/d_0}$. Using (17), this ensures that $\mu(B(x, r)) \geq 2k/m$. Further taking

$$r' = \sqrt{\Delta^2 + \frac{\rho + \Delta}{\rho} r^2}$$

$$r'' = \sqrt{\frac{\rho}{\rho - \Delta}((r')^2 - \Delta^2)} = \sqrt{\frac{\rho + \Delta}{\rho - \Delta}} \left(\frac{2k}{c_1 m}\right)^{1/d_0}$$

and using (21) and (22), we have, $B_M(x, r) \subset B_M(\theta, r') \subset B_M(x, r'')$. Since $\mu(B_M(x, r)) \geq 2k/m$, this ensures that $C(\theta) \subset B_M(x, r')$ which in turn is contained in $B_M(x, r'')$. Since for any good $\theta$, $\Delta \leq \rho/2$, the result follows. $\square$

### F.3 Proof of Lemma B.4

*Proof.* Let $E_1$ be the event that bounds the diameter of the response region of any good $\theta$, as specified in Lemma B.3, with probability at least $1 - \delta$, over the random choice of the training set. We condition on this event. For any $x \in M$ and $r > 0$, let

$$A(x, r) = \{\theta \in \Gamma_{\rho/2} : \|\pi_M(\theta) - x\| < r\}$$

Since $\frac{1}{2} < \sqrt{\frac{\rho - \Delta}{\rho + \Delta}}$ for $\Delta \leq \rho/2$, using Lemma B.3 and choosing large enough $m$ satisfying $(2k/(c_1 m))^{1/d_0} < \min(\rho, r_0)$, if we set

$$r_1 = \frac{1}{2} \left(\frac{k}{2c_2 m}\right)^{1/d_0}$$

then,

$$\theta \in A(x, r_1) \Rightarrow x \in B_M \left(\pi_M(\theta), \frac{1}{2} \left(\frac{k}{2c_2 m}\right)^{1/d_0}\right) \Rightarrow x \in B_M \left(\pi_M(\theta), \sqrt{\frac{\rho - \Delta}{\rho + \Delta}} \left(\frac{k}{2c_2 m}\right)^{1/d_0}\right)$$

which implies $\Rightarrow x \in C(\theta)$. It has been established that $A(x, r)$ has non-negligible probability mass under $\nu$.

**Lemma F.3** (Lemma 14 of Dasgupta and Tosh [2020]). *Suppose $\nu$ is multivariate Gaussian $N(0, I)$. There is a constant $c_d$, that depends on the dimension $d$, such that any $x \in M$ and $0 < r < r_0$, we have $\nu(A(x, r)) \geq c_d r^{d_0}$.*

Using (17), $M$ has a $r_1/2$ cover $\hat{M}$ of size at most $(2c_2/c_1) 8^{d_0} m/k$. To see this pick points $x_1, \ldots, x_n \in M$ that are at a distance $r_1/2$ from each other. The balls $B(x_1, r_1/4)$ are disjoint and each has $\mu(B(x_i, r_1/4)) \geq c_1(r_1/4)^{d_0} = (c_1/(2c_2))(1/8^{d_0})k/m$. Since total probability mass of these $N$ balls is at most 1, this gives the bound on $N$.

Pick any $\hat{x} \in \hat{M}$. For $i \in [m]$, let $U_i$ be a binary random variable that takes value 1 if $\theta_i \in A(\hat{x}, r_1/2)$ and 0 otherwise. Therefore,

$$U_i = \begin{cases} 1 & \text{w.p. } \nu(A(\hat{x}, r_1/2)) \\ 0 & \text{w.p.} 1 - \nu(A(\hat{x}, r_1/2)) \end{cases}$$

Note that $\nu(\hat{x}, r_1/2) \geq c_d(r_1/2)^{d_0} = (c_d/(2c_2))(1/4^{d_0})k/m = \alpha_d k/m$, where we have set $\alpha_d = (c_d/(2c_2))(1/4^{d_0})$. Let $U = \sum_{i=1}^m U_i$ be that number of $\theta_i$'s that fall in $A(\hat{x}, r_1/2)$ and $\mathbb{E}U =$

$\sum_{i=1}^{m} \mathbb{E}U_i \geq \alpha_d k$. Let $E_2$ be the event that $U > \alpha_d k/2$ and let $E_2^c$ be the complement event. Using Chernoff bound, we have

$$\Pr(E_2^c|E_1) = \Pr\left(U \leq \alpha_d k/2|E_1\right) \leq \Pr\left(U \leq (1/2)\mathbb{E}U|E_1\right) \leq e^{-\mathbb{E}U/8} \leq e^{-\alpha_d k/8}$$

Bounding the right most quantity above to be at most $\delta/\hat{M}$, for $k$ as specified in the lemma statement, for suitable choice of $c_d'$, we conclude that $E_2$ holds (conditioned on $E_1$) with probability at least $1 - \delta$. Therefore, with probability at least $1 - 2\delta$, both $E_1$ and $E_2$ hold, impying that for every $\hat{x} \in \hat{M}$, there are at least $\alpha_d k/2$ good $\theta_i$'s in $A(\hat{x}, r_1/2)$.

Now pick any arbitrary $x \in M$. There is some $\hat{x} \in \hat{M}$ with $\|x - \hat{x}\| \leq r_1/2$. Moreover, for any $\theta_j \in A(\hat{x}, r_1/2) \Rightarrow \theta_j \in A(x, r_1/2) \Rightarrow x \in C(\theta_j)$. $\qquad\square$

### F.4  Proof of Lemma B.1

For ease of exposition, we use the notation $x_{[1,n]}$ to denote $x_1, \ldots, x_n$ and $y_{[1,n]}$ to denote $y_1, \ldots, y_n$, $x'_{[1,n]}$ to denote $x'_1, \ldots, x'_n$ and $y'_{[1,n]}$ to denote $y'_1, \ldots, y'_n$ for various proofs appearing in the section. Note number of non-zero entries in the EaS representation given in (14) is variable and we use the notation $k(x)$ to denote the number of non-zero entries in $h_2(x)$.

*Proof.* Fix any $\Theta$. Then, conditioning on $x'_{[1,n]}$ ensures that $C_1, \ldots, C_m$ are fixed. Now,

$$\mathbb{E}_{x,D_n,D'_n}(\hat{\eta}(x) - \bar{\eta}(x))^2$$

$$= \mathbb{E}_{x,x_{[1,n]},y_{[1,n]},x'_{[1,n]},y'_{[1,n]}}\left[(\hat{\eta}(x) - \bar{\eta}(x))^2\right]$$

$$\overset{a}{=} \mathbb{E}_{x,x_{[1,n]},y_{[1,n]},x'_{[1,n]}}\left[(\hat{\eta}(x) - \bar{\eta}(x))^2\right]$$

$$= \mathbb{E}_{x_{[1,n]},x'_{[1,n]}}\left[\mathbb{E}_{x,y_{[1,n]}}\left[(\hat{\eta}(x) - \bar{\eta}(x))^2\Big|x_{[1,n]},x'_{[1,n]}\right)\right]\right]$$

$$= \mathbb{E}_{x_{[1,n]},x'_{[1,n]}}\left[\mathbb{E}_{x,y_{[1,n]}}\left[\left(\frac{1}{t}\sum_{j:x\in C_j}(\hat{\eta}_j - \bar{\eta}_j)\right)^2\Big|x_{[1,n]},x'_{[1,n]}\right]\right]$$

$$\overset{b}{\leq} \mathbb{E}_{x_{[1,n]},x'_{[1,n]}}\left[\mathbb{E}_{x,y_{[1,n]}}\left[\frac{1}{t}\sum_{j:x\in C_j}(\hat{\eta}_j - \bar{\eta}_j)^2\Big|x_{[1,n]},x'_{[1,n]}\right]\right]$$

$$= \mathbb{E}_{x_{[1,n]},x'_{[1,n]}}\left[\mathbb{E}_{x,y_{[1,n]}}\left[\frac{1}{t}\sum_{j:x\in C_j}\left(\frac{\sum_{i=1}^n y_i \mathbb{I}[x_i \in C_j]}{\sum_{i=1}^n \mathbb{I}[x_i \in C_j]} - \eta(C_j)\right)^2 \mathbb{I}[j \in A_t(x)]\Big|x_{[1,n]},x'_{[1,n]}\right]\right]$$

$$= \mathbb{E}_{x_{[1,n]},x'_{[1,n]}}\left[\mathbb{E}_{x,y_{[1,n]}}\left[\frac{1}{t}\sum_{j:x\in C_j}\left(\frac{\sum_{i=1}^n y_i \mathbb{I}[x_i \in C_j]}{n\mu_n(C_j)} - \eta(C_j)\right)^2 \mathbb{I}[j \in A_t(x)]\Big|x_{[1,n]},x'_{[1,n]}\right]\right]$$

$$= \mathbb{E}_{x_{[1,n]},x'_{[1,n]}}\left[\mathbb{E}_{x,y_{[1,n]}}\left[\frac{1}{t}\sum_{j:x\in C_j}\left(\left(\frac{\sum_{i=1}^n y_i \mathbb{I}[X_i \in C_j]}{n\mu_n(C_j)} - \eta(C_j)\right)^2 \mathbb{I}[n\mu_n(C_j) > 0]\mathbb{I}[j \in A_t(x)]\right.\right.\right.$$
$$\left.\left.\left. + \eta^2(C_j)\mathbb{I}[\mu_n(C_j) = 0]\mathbb{I}[j \in A_t(x)]\right)\Big|x_{[1,n]},x'_{[1,n]}\right]\right]$$

$$= \mathbb{E}_{x_{[1,n]},x'_{[1,n]}}\left[\mathbb{E}_{x,y_{[1,n]}}\left[\frac{1}{t}\sum_{j:x\in C_j}\left(\frac{\sum_{i=1}^n(y_i - \eta(C_j))\mathbb{I}_{\{x_i\in C_j\}}}{n\mu_n(C_j)}\right)^2 \mathbb{I}[n\mu_n(C_j) > 0]\mathbb{I}[j \in A_t(x)]\Big|x_{[1,n]},x'_{[1,n]}\right]\right]$$

$$+ \mathbb{E}_{x_{[1,n]},x'_{[1,n]}}\left[\mathbb{E}_{x,y_{[1,n]}}\left[\frac{1}{t}\sum_{j:x\in C_j}\eta^2(C_j)\mathbb{I}[\mu_n(C_j) = 0]\mathbb{I}[j \in A_t(x)]\Big|x_{[1,n]},x'_{[1,n]}\right]\right]$$

$$\overset{c}{\leq} \mathbb{E}_{x_{[1,n]},x'_{[1,n]}}\left[\mathbb{E}_{x,y_{[1,n]}}\left[\frac{1}{t}\sum_{j:x\in C_j}\left(\frac{\sum_{i=1}^n(y_i - \eta(C_j))\mathbb{I}[x_i \in C_j]}{n\mu_n(C_j)}\right)^2 \mathbb{I}[n\mu_n(C_j) > 0]\mathbb{I}[j \in A_t(x)]\Big|x_{[1,n]},x'_{[1,n]}\right]\right]$$
$$+ \frac{m}{tne}$$

$$= \mathbb{E}_{x_{[1,n]},x'_{[1,n]}}\left[\mathbb{E}_x\left[\frac{1}{t}\mathbb{E}_{y_{[1,n]}}\left[\sum_{j:x\in C_j}\left(\frac{\sum_{i=1}^n(y_i - \eta(C_j))\mathbb{I}[x_i \in C_j]}{n\mu_n(C_j)}\right)^2 \mathbb{I}[n\mu_n(C_j) > 0]\mathbb{I}[j \in A_t(x)]\right]\Big|x,x_{[1,n]},x'_{[1,n]})\right]\right] + \frac{m}{tne}$$

$$\overset{d}{\leq} \frac{1}{4}\mathbb{E}_{x_{[1,n]},x'_{[1,n]}}\left[\mathbb{E}_x\left(\frac{1}{t}\sum_{j:x\in C_j}\frac{\mathbb{I}[n\mu_n(C_j) > 0]\mathbb{I}[j \in A_t^x]}{n\mu_n(C_j)}\Big|x_{[1,n]},x'_{[1,n]}\right)\right] + \frac{m}{tne}$$

$$\overset{e}{\leq} \frac{1}{4}\left(\frac{2m}{t(n+1)}\right) + \frac{m}{tne} = \frac{m}{2t(n+1)} + \frac{m}{kne} \leq \frac{m}{2tn} + \frac{m}{tne} \leq \frac{m}{tn}.$$

where equality $a$ follows from the fact that the quantity with the inner square bracket is unaffected by the $y_i$'s. Inequality $b$ is due to Jensen's inequality. Inequality $c$ follows from Lemma F.4, in equality $d$

follows from Lemma F.6 and inequality $e$ follows from Lemma F.7. The result finally follows noting that by Jensen's inequality $\mathbb{E}|\hat{\eta}(x) - \bar{\eta}(x)| \leq \sqrt{\mathbb{E}(\hat{\eta}(x) - \bar{\eta}(x))^2}$. $\qquad\square$

**Lemma F.4.** *Pick $m \times d$ projection matrix $\Theta$. Suppose the `EaS` representation uses (i) a mapping $\Theta$ and (ii) empirical $k$-threshold sparsification. Let $x$ be sampled from $\mu$ and let $D_n = ((x_1, y_1), \ldots, (x_n, y_n))$ is a random training set where $x_i$ is sampled from $\mu$ and $y_i$ is distributed as $\eta(x_i)$ for $i \in [n]$. Similarly, $D'_n = ((x'_1, y'_1), \ldots, (x'_n, y'_n))$ be another random training set independent of $D_n$, where $x'_i$ is sampled from $\mu$ and $y'_i$ is distributed as $\eta(x'_i)$ for $i \in [n]$. Then the following holds.*

$$\mathbb{E}_{x_{[1,n]}, x'_{[1,n]}} \left[ \mathbb{E}_{x, y_{[1,n]}} \left[ \frac{1}{t} \sum_{j:x \in C_j} \eta^2(C_j) \mathbb{I}[\mu_n(C_j) = 0] \mathbb{I}[j \in A_t(x)] \Big| x_{[1,n]}, x'_{[1,n]} \right] \right] \leq \frac{m}{nte}$$

*Proof.* We start by taking the $1/t$ factor outside.

$$\frac{1}{t} \mathbb{E}_{x_{[1,n]}, x'_{[1,n]}} \left[ \mathbb{E}_{x, y_{[1,n]}} \left[ \sum_{j:x \in C_j} \eta^2(C_j) \mathbb{I}[\mu_n(C_j) = 0] \mathbb{I}[j \in A_t(x)] \Big| x_{[1,n]}, x'_{[1,n]} \right] \right]$$

$$\overset{a}{=} \frac{1}{t} \mathbb{E}_{x_{[1,n]}, x'_{[1,n]}} \left[ \mathbb{E}_x \left[ \sum_{j:x \in C_j} \eta^2(C_j) \mathbb{I}[\mu_n(C_j) = 0] \mathbb{I}[j \in A_t(x)] \Big| x_{[1,n]}, x'_{[1,n]} \right] \right]$$

$$= \frac{1}{t} \mathbb{E}_{x_{[1,n]}, x'_{[1,n]}} \left[ \sum_{k'=0}^{m} \Pr(k(x) = k') \mathbb{E}_x \left[ \sum_{j:x \in C_j} \eta^2(C_j) \mathbb{I}[n\mu_n(C_j) = 0] \mathbb{I}[j \in A_t(x)] \Big| k(x) = k' \right] \Big| x_{[1,n]}, x'_{[1,n]} \right]$$

$$\overset{b}{=} \frac{1}{t} \mathbb{E}_{x_{[1,n]}, x'_{[1,n]}} \left[ \sum_{k'=1}^{m} \Pr(k(x) = k') \mathbb{E}_x \left[ \sum_{j:x \in C_j} \eta^2(C_j) \mathbb{I}[n\mu_n(C_j) = 0] \mathbb{I}[j \in A_t(x)] \Big| k(x) = k' \right] \Big| x_{[1,n]}, x'_{[1,n]} \right]$$

$$\overset{c}{\leq} \frac{1}{t} \mathbb{E}_{x_{[1,n]}, x'_{[1,n]}} \left[ \sum_{k'=1}^{m} \Pr(k(x) = k') \mathbb{E}_x \left[ \sum_{j:x \in C_j} \mathbb{I}[n\mu_n(C_j) = 0] \mathbb{I}[j \in A_t(x)] \Big| k(x) = k' \right] \Big| x_{[1,n]}, x'_{[1,n]} \right]$$

$$\overset{d}{\leq} \frac{1}{t} \mathbb{E}_{x_{[1,n]}, x'_{[1,n]}} \left[ \sum_{k'=1}^{m} \Pr(k(x) = k') \sum_{j=1}^{m} \mu(C_j) \mathbb{I}[n\mu_n(C_j) = 0] \Big| x_{[1,n]}, x'_{[1,n]} \right]$$

$$\overset{e}{\leq} \frac{1}{t} \mathbb{E}_{x_{[1,n]}, x'_{[1,n]}} \left[ \Pr(k(x) = 0) \sum_{j=1}^{m} \mu(C_j) \mathbb{I}[n\mu_n(C_j)] + \sum_{k'=1}^{m} \Pr(k(x) = k') \sum_{j=1}^{m} \mu(C_j) \mathbb{I}[n\mu_n(C_j) = 0] \Big| x_{[1,n]}, x'_{[1,n]} \right]$$

$$= \frac{1}{t} \mathbb{E}_{x_{[1,n]}, x'_{[1,n]}} \left[ \sum_{j=1}^{m} \mu(C_j) \mathbb{I}[n\mu_n(C_j) = 0] \left( \sum_{k'=0}^{m} \Pr(k(x) = k') \right) \Big| x_{[1,n]}, x'_{[1,n]} \right]$$

$$= \frac{1}{t} \mathbb{E}_{x_{[1,n]}, x'_{[1,n]}} \left[ \sum_{j=1}^{m} \mu(C_j) \mathbb{I}[n\mu_n(C_j) = 0] \Big| x_{[1,n]}, x'_{[1,n]} \right] = \frac{1}{t} \mathbb{E}_{x'_{[1,n]}} \left[ \sum_{j=1}^{m} \mathbb{E}_{x_{[1,n]}} \left[ \mu(C_j) \mathbb{I}[n\mu_n(C_j) = 0] \Big| x'_{[1,n]} \right] \right]$$

$$\leq \frac{1}{t} \mathbb{E}_{x'_{[1,n]}} \left[ \sum_{j=1}^{m} \mu(C_j)(1 - \mu(C_j))^n \right] = \frac{1}{nt} \mathbb{E}_{x'_{[1,n]}} \left[ \sum_{j=1}^{m} n\mu(C_j)(1 - \mu(C_j))^n \right]$$

$$\leq \frac{1}{nt} \mathbb{E}_{x'_{[1,n]}} \left[ \sum_{j=1}^{m} n\mu(C_j) e^{-n\mu(C_j)} \right] \leq \frac{m}{nt} \mathbb{E}_{x'_{[1,n]}} \left[ \max_j \left\{ n\mu(C_j) e^{-n\mu(C_j)} \right\} \right] \overset{f}{\leq} \frac{m}{nt} \mathbb{E}_{x'_{[1,n]}} \left[ \frac{1}{e} \right] = \frac{m}{nte}$$

where equality $a$ follows from the fact that the quantity with the inner square bracket is unaffected by the $y_i$'s. Equality $b$ follows from the fact that for $k(x) = 0$, number of non-zero entries in the `EaS` of $x$ using (14) is zero and $x \notin C_j, \forall j$. Therefore, we denote the quantity

$\sum_{j:x\in C_j} \eta^2(C_j)\mathbb{I}[n\mu_n(C_j)]\mathbb{I}[j \in A_t^x] = 0$. Inequality $c$ follows from the fact that $\eta(C_j) \leq 1, \forall j$. Inequality $d$ follows from Lemma F.5. Inequality $e$ follows since we are adding a non-negative quantity. Finally, inequality $f$ follows from the fact that $\sup_z ze^{-z} = \frac{1}{e}$. $\qquad\square$

**Lemma F.5.**

$$\mathbb{E}_x \left[ \sum_{j:x\in C_j} \mathbb{I}[n\mu_n(C_j) = 0]\mathbb{I}[j \in A_t(x)] \Big| k(x) = k' \right] \leq \sum_{j=1}^{m} \mu(C_j)\mathbb{I}[n\mu_n(C_j) = 0]$$

*Proof.* Conditioned on $k(x) = k'$, `EaS` representation given in (14) has exactly $k'$ non-zero entries. Therefore, using Lemma 3.4,

$$\mathbb{E}_x \left[ \sum_{j:x\in C_j} \mathbb{I}[n\mu_n(C_j) = 0]\mathbb{I}[j \in A_t(x)] \Big| k(x) = k' \right]$$

$$\stackrel{a}{=} \sum_{i=1}^{\binom{m}{k'}} \Pr\left( x \in C_i^{k'} \right) \sum_{j:\sigma(i)} \mathbb{I}[n\mu_n(C_j) = 0]\mathbb{I}[j \in A_t(x)]$$

$$= \sum_{i=1}^{\binom{m}{k'}} \sum_{j:\sigma(i)} \mu(C_i^{k'})\mathbb{I}[n\mu_n(C_j) = 0]\mathbb{I}[j \in A_t(x)]$$

$$\stackrel{b}{=} \sum_{j=1}^{m} \sum_{i:j\in\sigma(i)} \mu(C_i^{k'})\mathbb{I}[n\mu_n(C_j) = 0]\mathbb{I}[j \in A_t(x)]$$

$$= \sum_{j=1}^{m} \mathbb{I}[n\mu_n(C_j) = 0] \left( \sum_{i:j\in\sigma(i)} \mu(C_i^{k'})\mathbb{I}[j \in A_t(x)] \right)$$

$$\leq \sum_{j=1}^{m} \mathbb{I}[n\mu_n(C_j) = 0] \left( \sum_{i:j\in\sigma(i)} \mu(C_i^{k'}) \right) \stackrel{c}{=} \sum_{j=1}^{m} \mu(C_j)\mathbb{I}[n\mu_n(C_j) = 0]$$

where equality $a$ follows from part (ii) of lemma 3.4 since $\{C_i^{k'}\}_{i=1}^{\binom{m}{k'}}$ forms a partition of $\mathcal{X}$ and the definition of $\sigma$ in section 3.2 with $k = k'$. Equality $b$ follows from the following observation. In the line above inequality $b$, we are summing $k' \times \binom{m}{k'}$ terms. Since $k' \times \binom{m}{k'} = m \times \binom{m-1}{k'-1}$ and any $j \in [m]$ can appear in exactly $\binom{m-1}{k'-1}$ different subsets of of size $k'$, equality $b$ is simply rearranging the terms from the line above by changing the indices appropriately. Equality $c$ follows from part (iv) of lemma 3.4. $\qquad\square$

**Lemma F.6.** *Pick $m \times d$ projection matrix $\Theta$. Suppose the `EaS` representation uses (i) a mapping $\Theta$ and (ii) empirical k-threshold sparsification. Let $x$ be sampled from $\mu$ and let $D_n = ((x_1, y_1), \ldots, (x_n, y_n))$ is a random training set where $x_i$ is sampled from $\mu$ and $y_i$ is distributed as $\eta(x_i)$ for $i \in [n]$. Similarly, $D_n' = ((x_1', y_1'), \ldots, (x_n', y_n'))$ be another random training set independent of $D_n$, where $x_i'$ is sampled from $\mu$ and $y_i'$ is distributed as $\eta(x_i')$ for $i \in [n]$. Then the following holds.*

$$\mathbb{E}_{y_{[1,n]}} \left[ \sum_{j:x\in C_j} \left( \frac{\sum_{i=1}^{n}(y_i - \eta(C_j))\mathbb{I}[x_i \in C_j]}{n\mu_n(C_j)} \right)^2 \mathbb{I}[n\mu_n(C_j) > 0]\mathbb{I}[j \in A_t(x)] \Big| x, x_{[1,n]}, x_{[1,n]}' \right]$$

$$\leq \frac{1}{4} \sum_{j:x\in C_j} \left( \frac{\mathbb{I}[n\mu_n(C_j) > 0]\mathbb{I}[j \in A_t(x)]}{n\mu_n(C_j)} \right)$$

*Proof.* Conditioned on $x$ and $x_{[1,n]}'$, only $k(x) = k'$ of the $m$ coordinates in the `EaS` representation of $x$ are non-zero. If $k' = 0$ then the right hand side of the Lemma statement still holds with tghe value being zero since each $C_j$ will be an empty set in that case. WLOG, for ease of exposition,

assume that $k' > 0$ and these $k'$ non-zero coordinate to be $j_1, \ldots, j_{k'} \in [m]$. Then the number of $x_i$ that falls in any such $C_{j_l}$, where $l \in [k']$, is $n\mu_n(C_{j_l})$. The $y_i$ values corresponding to these $x_i$ points (there are $n\mu_n(C_{j_l})$ of them in total) are identically and independently distributed with expectation

$$
\begin{aligned}
\mathbb{E}(y_i | x_i \in C_{j_l}) &= \Pr(y_i = 1 | x_i \in C_{j_l}) = \frac{1}{\mu(C_{j_l})} \int_{C_{j_l}} \Pr(y_i = 1 | x_i = x) \mu(dx) \\
&= \frac{1}{\mu(C_{j_l})} \int_{C_{j_l}} \eta(x)\mu(dx) = \eta(C_{j_l})
\end{aligned}
$$

Therefore, we can write

$$
\mathbb{E}_{y_{[1,n]}} \left[ \sum_{j:x\in C_j} \left( \frac{\sum_{i=1}^n (y_i - \eta(C_j))\mathbb{I}[x_i \in C_j]}{n\mu_n(C_j)} \right)^2 \mathbb{I}[n\mu_n(C_j) > 0]\mathbb{I}[j \in A_t(x)] \bigg| x, x_{[1,n]}, x'_{[1,n]} \right]
$$

$$
= \sum_{l=1}^{k'} \left[ \frac{\mathbb{E}_{y_{[1,n]}} \left[ \left(\sum_{i=1}^n (y_i - \eta(C_{j_l}))\mathbb{I}[x_i \in C_{j_l}]\right)^2 \mathbb{I}[n\mu_n(C_{j_l}) > 0]\mathbb{I}[j_l \in A_t(x)] \bigg| x, x_{[1,n]}, x'_{[1,n]} \right]}{(n\mu_n(C_{j_l}))^2} \right]
$$

$$
\stackrel{a}{=} \sum_{l=1}^{k'} \left[ \frac{\sum_{i=1}^n \mathbb{E}_{y_i} \left[ (y_i - \eta(C_{j_l}))^2 \mathbb{I}[x_i \in C_{j_l}]\mathbb{I}[n\mu_n(C_{j_l}) > 0]\mathbb{I}[j_l \in A_t(x)] \bigg| x, x_i, x'_i \right]}{(n\mu_n(C_{j_l}))^2} \right]
$$

$$
\stackrel{b}{=} \sum_{l=1}^{k'} \left[ \frac{\sum_{i=1}^n \eta(C_{j_l})(1 - \eta(C_{j_l}))\mathbb{I}[x_i \in C_{j_l}]\mathbb{I}[n\mu_n(C_{j_l}) > 0]\mathbb{I}[j_l \in A_t(x)]}{(n\mu_n(C_{j_l}))^2} \right]
$$

$$
\stackrel{c}{\le} \frac{1}{4} \sum_{l=1}^{k} \left( \frac{\mathbb{I}[n\mu_n(C_{j_l}) > 0]\mathbb{I}[j_l \in A_t(x)]}{n\mu_n(C_{j_l})} \right)
$$

where, equality $a$ is due to the following observation. For any $i, j \in [m], i \ne j$ and $x_i, x_j \in C_l$ for some $l \in [m]$, $y_i$ and $y_j$ are identically and independently distributed with expectation $\eta(C_l)$. Therefore, the expectation of the cross product is simply:

$$
\begin{aligned}
\mathbb{E}_{y_i,y_j} \left[ (y_i - \eta(C_l))(y_j - \eta(C_l)) \right] &= \mathbb{E}_{y_i,y_j} \left[ y_iy_j - \eta(C_l)(y_i + y_j) + \eta(C_l)^2 \right] \\
&= \mathbb{E}y_i\mathbb{E}y_j - \eta(C_l)(\mathbb{E}y_i + \mathbb{E}y_j) + \eta(C_l)^2 = 0.
\end{aligned}
$$

Equality $b$ follows from variance computation. In particular for any $y_i, i \in [m]$ with $x_i \in C_l$ for some $l \in [m]$,

$$
\mathbb{E}\left[ (y_i - \eta(C_l))^2 \right] = \mathbb{E}y_i^2 - 2\eta(C_l)\mathbb{E}y_i + \eta(C_l)^2 = \eta(C_l) - 2\eta(C_l)^2 + \eta(C_l)^2 = \eta(C_l)(1 - \eta(C_l)).
$$

Finally, inequality $c$ follows from the fact that for any $z \in [0, 1]$, the maximum value of $z(1 - z)$ is $\frac{1}{4}$.

It is easy to observe that the final result is equivalent to $\frac{1}{4} \sum_{j:x\in C_j} \left( \frac{\mathbb{I}[n\mu_n(C_j)>0]\mathbb{I}[j\in A_t(x)]}{n\mu_n(C_j)} \right)$. $\qquad \square$

**Lemma F.7.** *Pick $m \times d$ projection matrix $\Theta$. Suppose the EaS representation uses (i) a mapping $\Theta$ and (ii) empirical $k$-threshold sparsification. Let $x$ be sampled from $\mu$ and let $D_n = ((x_1, y_1), \ldots, (x_n, y_n))$ is a random training set where $x_i$ is sampled from $\mu$ and $y_i$ is distributed as $\eta(x_i)$ for $i \in [n]$. Similarly, $D'_n = ((x'_1, y'_1), \ldots, (x'_n, y'_n))$ be another random training set independent of $D_n$, where $x'_i$ is sampled from $\mu$ and $y'_i$ is distributed as $\eta(x'_i)$ for $i \in [n]$. Then the following holds.*

$$
\mathbb{E}_{x_{[1,n]},x'_{[1,n]}} \left[ \mathbb{E}_x \left[ \frac{1}{t} \sum_{j:x\in C_j} \frac{\mathbb{I}[\mu_n(C_j) > 0]\mathbb{I}[j \in A_t(x)]}{n\mu_n(C_j)} \bigg| x_{[1,n]}, x'_{[1,n]} \right] \right] \le \frac{2m}{t(n+1)}
$$

*Proof.* We start by taking the $1/t$ factor outside.

$$\frac{1}{t}\mathbb{E}_{x_{[1,n]},x'_{[1,n]}}\left[\mathbb{E}_x\left[\sum_{j:x\in C_j}\frac{\mathbb{I}[\mu_n(C_j)>0]\mathbb{I}[j\in A_t(x)]}{n\mu_n(C_j)}\Big|x_{[1,n]},x'_{[1,n]}\right]\right]$$

$$=\frac{1}{t}\mathbb{E}_{x_{[1,n]},x'_{[1,n]}}\left[\sum_{k'=0}^{m}\Pr(k(x)=k')\mathbb{E}_x\left[\sum_{j:x\in C_j}\frac{\mathbb{I}[n\mu_n(C_j)>0]\mathbb{I}[j\in A_t(x)]}{n\mu_n(C_j)}\Big|k(x)=k'\right]\Big|x_{[1,n]},x'_{[1,n]}\right]$$

$$\overset{a}{=}\frac{1}{t}\mathbb{E}_{x_{[1,n]},x'_{[1,n]}}\left[\sum_{k'=1}^{m}\Pr(k(x)=k')\mathbb{E}_x\left[\sum_{j:x\in C_j}\frac{\mathbb{I}[n\mu_n(C_j)>0]\mathbb{I}[j\in A_t(x)]}{n\mu_n(C_j)}\Big|k(x)=k'\right]\Big|x_{[1,n]},x'_{[1,n]}\right]$$

$$\overset{b}{\le}\frac{1}{t}\mathbb{E}_{x_{[1,n]},x'_{[1,n]}}\left[\sum_{k'=1}^{m}\Pr(k(x)=k')\sum_{j=1}^{m}\frac{\mu(C_j)\mathbb{I}[n\mu_n(C_j)>0]}{n\mu_n(C_j)}\Big|x_{[1,n]},x'_{[1,n]}\right]$$

$$\overset{c}{\le}\frac{1}{t}\mathbb{E}_{x_{[1,n]},x'_{[1,n]}}\left[\Pr(k(x)=0)\sum_{j=1}^{m}\frac{\mu(C_j)\mathbb{I}[n\mu_n(C_j)>0]}{n\mu_n(C_j)}+\sum_{k'=1}^{m}\Pr(k(x)=k')\sum_{j=1}^{m}\frac{\mu(C_j)\mathbb{I}[n\mu_n(C_j)=0]}{n\mu_n(C_j)}\Big|x_{[1,n]},x'_{[1,n]}\right]$$

$$=\frac{1}{t}\mathbb{E}_{x_{[1,n]},x'_{[1,n]}}\left[\sum_{j=1}^{m}\frac{\mu(C_j)\mathbb{I}[n\mu_n(C_j)>0]}{n\mu_n(C_j)}\left(\sum_{k'=0}^{m}\Pr(k(x)=k')\right)\Big|x_{[1,n]},x'_{[1,n]}\right]$$

$$=\frac{1}{t}\mathbb{E}_{x_{[1,n]},x'_{[1,n]}}\left[\sum_{j=1}^{m}\frac{\mu(C_j)\mathbb{I}[n\mu_n(C_j)>0]}{n\mu_n(C_j)}\Big|x_{[1,n]},x'_{[1,n]}\right]$$

$$=\frac{1}{t}\mathbb{E}_{x'_{[1,n]}}\left[\sum_{j=1}^{m}\mathbb{E}_{x_{[1,n]}}\left[\frac{\mu(C_j)\mathbb{I}[n\mu_n(C_j)>0]}{n\mu_n(C_j)}\Big|x'_{[1,n]}\right]\right]$$

$$\overset{d}{\le}\frac{1}{t}\mathbb{E}_{x'_{[1,n]}}\left[\sum_{j=1}^{m}\frac{2\mu(C_j)}{(n+1)\mu(C_j)}\right]=\frac{1}{t}\mathbb{E}_{x'_{[1,n]}}\left[\frac{2m}{(n+1)}\right]=\frac{2m}{t(n+1)}$$

where equality $a$ follows from the fact that for $k(x)=0$, number of non-zero entries in the EaS of $x$ using (14) is zero and $x\notin C_j,\forall j$. Therefore, we denote the quantity $\sum_{j:x\in C_j}\frac{\mathbb{I}[n\mu_n(C_j)>0]\mathbb{I}[j\in A_t(x)]}{n\mu_n(C_j)}=0$. Inequality $b$ follows from Lemma F.8. Inequality $c$ follows since we are adding a non-negative quantity. Finally, inequality $d$ follows from the fact that $n\mu_n(C_j)$ is Binomially distributed with parameters $n$ and $\mu(C_j)$ and by an application of lemma E.4. $\square$

**Lemma F.8.**

$$\mathbb{E}_x\left[\sum_{j:x\in C_j}\frac{\mathbb{I}[n\mu_n(C_j)>0]\mathbb{I}[j\in A_t(x)]}{n\mu_n(C_j)}\Big|k(x)=k'\right]\le\sum_{j=1}^{m}\frac{\mu(C_j)\mathbb{I}[n\mu_n(C_j)>0]}{n\mu_n(C_j)}$$

*Proof.* Conditioned on $k(x) = k'$, EaS representation given in (14) has exactly $k'$ non-zero entries. Therefore, using Lemma 3.4,

$$\mathbb{E}_x \left[ \sum_{j:x \in C_j} \frac{\mathbb{I}[n\mu_n(C_j) > 0]\mathbb{I}[j \in A_t(x)]}{n\mu_n(C_j)} \middle| k(x) = k' \right]$$

$$\stackrel{a}{=} \sum_{i=1}^{\binom{m}{k'}} \Pr\left( x \in C_i^{k'} \right) \sum_{j:\sigma(i)} \frac{\mathbb{I}[n\mu_n(C_j) > 0]\mathbb{I}[j \in A_t(x)]}{n\mu_n(C_j)}$$

$$= \sum_{i=1}^{\binom{m}{k'}} \sum_{j:\sigma(i)} \frac{\mu(C_i^{k'})\mathbb{I}[n\mu_n(C_j) > 0]\mathbb{I}[j \in A_t(x)]}{n\mu_n(C_j)}$$

$$\stackrel{b}{=} \sum_{j=1}^{m} \sum_{i:j \in \sigma(i)} \frac{\mu(C_i^{k'})\mathbb{I}[n\mu_n(C_j) > 0]\mathbb{I}[j \in A_t(x)]}{n\mu_n(C_j)}$$

$$= \sum_{j=1}^{m} \frac{\mathbb{I}[n\mu_n(C_j) > 0]\mathbb{I}[j \in A_t(x)]}{n\mu_n(C_j)} \left( \sum_{i:j \in \sigma(i)} \mu(C_i^{k'}) \right)$$

$$\leq \sum_{j=1}^{m} \frac{\mathbb{I}[n\mu_n(C_j) > 0]}{n\mu_n(C_j)} \left( \sum_{i:j \in \sigma(i)} \mu(C_i^{k'}) \right) \stackrel{c}{=} \sum_{j=1}^{m} \frac{\mu(C_j)\mathbb{I}[n\mu_n(C_j) > 0]}{n\mu_n(C_j)}$$

where equality $a$ follows from part (ii) of lemma 3.4 since $\{C_i^{k'}\}_{i=1}^{\binom{m}{k'}}$ forms a partition of $\mathcal{X}$ and the definition of $\sigma$ in section 3.2 with $k = k'$. Equality $b$ follows from the following observation. In the line above inequality $b$, we are summing $k' \times \binom{m}{k'}$ terms. Since $k' \times \binom{m}{k'} = m \times \binom{m-1}{k'-1}$ and any $j \in [m]$ can appear in exactly $\binom{m-1}{k'-1}$ different subsets of of size $k'$, equality $b$ is simply rearranging the terms from the line above by changing the indices appropriately. Equality $c$ follows from part (iv) of lemma 3.4. $\qquad\square$

