# OpenReview forum: "Non-parametric classification via expand-and-sparsify representation"
_NeurIPS.cc/2024/Conference — NeurIPS 2024 poster_

### Official Review · Reviewer_E4XV · 2024-06-27

**Soundness:** 3
**Presentation:** 3
**Contribution:** 3
**Rating:** 5
**Confidence:** 2

**Summary:**

This paper addresses how to use expand-and-sparsify techniques to non-parametric classification. More specifically, it uses expand-and-sparsify on the test data point $x$ and then use the response regions of $x$, which are the sets of training data sharing same activation coordinates as $x$, to serve as the neighbourhood of $x$ and then predict the label $y$. It proposes two types of sparsification algorithms for this problem. The first algorithm chooses the $k$ largest values of the higher dimension that $x$ maps to. The authors proves the convergence rate of this algorithm and shows that it is minimax-optimal. The second algorithm chooses the empirical $k$-thresholding for sparsification. Unlike the first algorithm, the authors assume the matrix $\Theta$ which maps $x$ to higher dimension to be multivaraite Gaussian. The authors then prove that the second algorithm is also minimax-optimal. Finally, the authors empirically show the performance of algorithm 1 and compare it with KNN and random forest under different dimensions ($m$).

**Strengths:**

1. The problem setting is novel as the authors combined the expand-and-sparsify technique to non-parametric classification.
2. The authors provide two different algorithms for sparsification and show that both of these are minimax-optimal.

**Weaknesses:**

1. typos: line 137, $\{(x_1, y_1)\}_{i=1}^n \to \{(x_i, y_i)\}_{i=1}^n$. line 348: should be \textit{performance of our proposed classifier}? line 380: Big Oh changes to Big $\mathcal{O}$?
2. The motivation and the use cases of the problem is not addressed, why is it necessary to use expand-and-sparsify in non-parametric classification and if it it used, in which scenarios will this help?
3. The datasets used in the experiments are not explained and there is no experiments for algorithm 2 and different values of $k$ for the sparsification.

**Questions:**

1. Is it possible to compare and contrast the two algorithms and specify which algorithm is better under different scenarios?
2. Why the performance of algorithm 1 in the experiments increases with m but the other methods are the same when m increases?

**Limitations:**

The motivation and applicability of the techniques used in the paper is not specified and the experiments only show the performance of the first algorithm.

---

> ### Author Rebuttal · Authors · 2024-08-05
>
> We thank the reviewer for providing constructive feedback and pointing out the typos (weakness 1). We will fix those.
>
> Regarding weakness 2, it is not ``necessary" to use expand-and-sparsify representation in non-parametric classification. Note that there different types of non-paranetric classifiers, such as,  histogram classifiers (partitioning estimates), $k$-nearest neighbor classifiers, kernel methods, random forest and so on. Our proposed classifier a new non-parametric classifier that utilizes expand-and-sparsify representation. We establish solid theoretical properties of our proposed classifier -- for appropriate choice of hyper-parameter, excess Bayes risk of our proposed classifier achieves minimax-optimal convergence rate (the best rate possible). Experimental results show a strong performance almost on par with the nearest neighbor and random forest classifiers.
>
> Regarding weakness 3, as mentioned in line 347, details of the datasets are provided in appendix A. As per Theorem 3.12, we use $k= d\log m$ in our experiments.
>
> To answer question 1, note that our Alg 1 achieves minimax-optimal convergence rate, the best possible in the non-parametric setting. This convergence rate decays exponentially slowly with (ambient) dimension $d$.  However, we have shown in section 3.5 that even if the data lie on a low dimensional manifold with intrinsic dimension $d_0\ll d$, there exists a smooth regression function $\eta$ such that the quantity $\mathbb{E}_{x,D_n}|\eta(x)-\hat{\eta(x)}|$, where $\hat{\eta}(x)$ is the conditional probability estimate of Alg 1, decreases at a rate no faster than $n^{-\frac{1}{d+1}}$. Therefore, we can not expect excess Bayes risk to decay (a bit faster) at a rate that decays exponentially in $d_0$, instead of $d$, which means Alg 1 can not adapts to the low dimensional manifold structure. Alg 2 is designed for this very purpose. When data lie on a low dimensional manifold with intrinsic dimension $d_0\ll d$, convergence rate of Alg 2 decays exponentially slowly in $d_0$ (instead of $d$) and is again minimax-optimal. Please also see our rebuttal to reviewer Q16i to have more discussion on this subject.
>
> To answer question 2, please note that $m$ is a hyper-parameter of our proposed Alg 1 and this hyper-parameter does not affect the performance of other non-parametric classifiers such as $k$-nearest neighbor ($k$-NN) classifier or random forest (RF) classifier. Our theoretical result (Theorem 3.12) suggests that excess Bayes risk decreases with increasing $m$. We wanted to verify this behavior in our empirical results. As a baseline, we chose two other non-parametric classifiers, namely $k$-NN amd RF. For $k$-NN, we chose two values of $k$, namely $k=1$ and $k=10$, which are pretty standard. For RF, we chose the number of trees via grid search and cross validation. Since $m$ is not a hyper-parameter of $k$-NN or RF, for different values of $m$, $k$-NN and RF test accuracies are constant.

---

> > ### Comment · Reviewer_E4XV · 2024-08-11
> >
> > Thanks for the authors' reply, the answers to questions 1 and 2 are pretty reasonable. I still feel the motivation is not clear for the algorithm and setting proposed by the paper, but I have bumped my score to 5 as the main contribution, as specified by the author, is to provide a novel framework and a theoretical guarantee.

---

> > > ### Author Response · Authors · 2024-08-11
> > >
> > > We thank the reviewer for acknowledging that our answers are reasonable. We apologize that the motivation of the algorithm and the setting are not clear. Let us try to argue about our motivation.
> > >
> > > In recent years, it has been observed that a striking neural architecture appears in the sensory systems of several living organisms, which essentially is a transformation from a low-dimensional dense representation of sensory stimulus to a much higher-dimensional, sparse representation. This has been found, for instance, in the olfactory system of the fly (Wilson [2013]) and mouse (Stettler and Axel [2009]), the visual system of the cat (Olshausen and Field [2004]), and the electrosensory system of the electric fish (Chacron et al. [2011]).
> > >
> > > For example, in the olfactory system of Drosophila  (Turner et al. [2008], Masse et al. [2009], Wilson [2013],  Caron et al. [2013]), the primary sense receptors of the fly are the roughly 2,500 odor receptor neurons (also known as, ORNs) in its antennae and maxillary palps, which can be clustered into 50 types, based on their odor responses, leading to  a dense, 50-dimensional sensory input vector. This information is then relayed via projection neurons to a collection of roughly 2000 Kenyon cells (KCs) in the mushroom body, with each KC receiving signal from roughly 5-10 glomeruli. The pattern of connectivity between the glomeruli and Kenyon cells appears random (Chacron et al. [2011]). The output of the KCs is integrated by a single anterior paired lateral (APL) neuron which then provides negative feedback causing all but the 5% highest-firing KCs to be suppressed (Lin et al. [2014]). The result is a *random* sparse high-dimensional representation of the sensory input, that is the basis for subsequent learning.
> > >
> > > The primary motivation of this paper is to study the benefit of this type of naturally occurring representation, namely expand-and-sparsify representation  (EaS for short) in the supervised classification setting. In our setting, given a dense vector $x\in \mathcal{S}^{d-1}$ on unit sphere, we first obtain high dimensional transformation $y=\Theta x \in \mathbb{R}^m$, where $d\ll m$, by multiplying $x$ with a random projection matrix $\Theta$. Following this, we obtain EaS representation of $x$, which is a $k$ sparse $m$ dimensional binary vector, by setting (activating) the largest $k$ entries of $y$ to 1 and the remaining entries to 0 (suppression). It turns out that all the examples $x\in\mathcal{S}^{d-1}$ that set the $j^{th}$ bit, for any $j=1,\ldots,m$, to 1 in their respective EaS representations can not be located too far. In fact, the diameter of the ball containing these points can be made small by appropriately choosing large $m$. Thus, we can estimate the average conditional probability of these regions using labeled data. Given any $x$, we can estimate the conditional probability $P(y=1|x)$ by taking the average of conditional probability estimates over $k$ regions corresponding to the $k$ non-zero bits in the EaS representation of $x$. Once we have such a conditional probability estimate, comparing it with $1/2$, we predict the class label in the binary classification setting. This is precisely the intuition of our algorithm 1. It turns out, that this proposed classifier admits the form of a locally weighted average classifier and can be analyzed using the ideas from non-parametric statistics as detailed in section 3. Alg 2 is a bit more complicated and is analyzed in section 4.
> > >
> > > **If the paper is accepted we will have one extra content page and if you are satisfied with the motivation and setting presented above, we can summarize the above discussion in the camera ready version.**
> > >
> > > Wilson [2013] : R. Wilson. Early olfactory processing in Drosophila: Mechanisms and principles. Annual Review of Neuroscience, 36:217–241, 2013.
> > >
> > > Stettler and Axel [2009]: D. Stettler and R. Axel. Representations of odor in the piriform cortex. Neuron, 63:854–864, 2009.
> > >
> > > Olshausen and Field [2004]: B. Olshausen and D. Field. Sparse coding of sensory inputs. Current Opinion in Neurobiology, 14:481–487, 2004.
> > >
> > > Chacron et al. [2011]: M. Chacron, A. Longtin, and L. Maler. Efficient computation via sparse coding in electrosensory neural networks. Current Opinion in Neurobiology, 21:752–760, 2011.
> > >
> > > Turner et al. [2008]: G. Turner, M. Bazhenov, and G. Laurent. Olfactory representations by Drosophila mushroom body neurons. J. Neurophysiol., 99:734–746, 2008.
> > >
> > > Masse et al. [2009]: N. Masse, G. Turner, and G. Jefferis. Olfactory information processing in Drosophila: review. Current Biology, 19:R700–R713, 2009.
> > >
> > > Caron et al. [2013]: S. Caron, V. Ruta, L. Abbott, and R. Axel. Random convergence of olfactory inputs in the Drosophila mushroom body. Nature, 497:113–117, 2013.
> > >
> > > Lin et al. [2014]: A. Lin, A. Bygrave, A. de Calignon, T. Lee, and G. Miesenbock. Sparse, decorrelated odor coding in the mushroom body enhances learned odor discrimination. Nature Neuroscience, 17(4), 2014.

---

> > > > ### Author Response · Authors · 2024-08-12
> > > > **Invitation for comments and clarifications**
> > > >
> > > > Dear Reviewer,
> > > >
> > > > We greatly value your feedback and have added motivation for the setting of our paper and algorithm 1. To ensure that we have properly addressed your concerns, we would greatly appreciate it if you could kindly review our responses and provide any further comments or questions you might have. We are looking forward to engaging with you before the discussion period ends.
> > > >
> > > > Thank you for your time and consideration.

---

### Official Review · Reviewer_4EZA · 2024-07-05

**Soundness:** 3
**Presentation:** 3
**Contribution:** 2
**Rating:** 6
**Confidence:** 3

**Summary:**

In this paper, the authors studied the problem of binary classification when the feature vectors are on the unit sphere in $\mathbb{R}^d$. The paper proposed a non-parametric algorithm based on a method called EaS (Expand and Sparsification). They proved that this method is universally consistent and that its errors converge to the Bayes error with a rate of $\mathcal{O}(n^{-\frac{1}{d}})$. They also proposed an algorithm for when the data lie in a $d_0$-dimensional manifold and proved that its errors converge to the Bayes error with a rate of $\mathcal{O}(n^{-\frac{1}{d_0}})$. The paper builds upon the work of Dasgupta and Tosh [2020], which is cited in the paper.

**Strengths:**

- The proofs are rigorous.

- The writing is very good and easy to follow.

- The messages of the paper are very clear.

**Weaknesses:**

- The contribution with respect to Dasgupta and Tosh [2020] is very marginal, and many of the ideas come from that work.

- In practice, other methods like random forests seem to perform better than the proposed method.

- It is not mentioned in the paper whether this is the best theoretical work with this rate of convergence.

- When data lie in a $d_0$-dimensional manifold, the convergence rate again seems exponentially slow with respect to $d$. Although this limitation is mentioned in the paper, I think it is a weakness.

**Questions:**

- I think in line $42$, $\mathbb{P}(y=1 \vert x)$ should be $\mathbb{P}(y=1 \vert C_j)$.

- In line $537$, what is $\nu$?

- The proofs in lines $536$ to $542$ are very unclear, and there are some claims in these lines which I think need mathematical proof.

- I think there are some inconsistencies in the notation of the proof of Lemma $3.5$ with the main body of the paper; please correct these.

**Limitations:**

The limitations are addressed in the paper.

---

> ### Author Rebuttal · Authors · 2024-08-05
>
> We thank the reviewer for providing constructive feedback.
>
> Regarding weakness 1, while our work is inspired by the work of Dasgupta and Tosh [2020], we have clearly articulated how our work is different and advances the result of Dasgupta and Tosh [2020] in lines 96-111.
>
> Regarding weakness 3, in the non-parametric setting, minimax-optimal convergence rate is the best possible convergence rate one can hope for. For appropriate choice of $m$, our proposed method achieves the minimax-optimal convergence rate (Remark 3.14, 4.4), therefore, this is the best possible rate.
>
> Regarding weakness 4, yes, this is a limitation of Alg 1 as Alg 1 does not adapt to low dimensional manifold. We have explicitly established a lower bound in Theorem 3.15. To address this shortcomings, we have developed Alg 2 whose convergence rate, when data lie on a low dimensional manifold with intrinsic dimension $d_0\ll d$, decays exponentially slowly in $d_0$ (Theorem 4.2, Corollary 4,3, Remark 4.3) which is minimax-optimal. It is well known (Tsybakov [2008], Gyorfi et al. [2002]) that, in the non-parametric setting, minimax-optimal convergence rate decays exponentially slowly with the dimension. When data lie on a low dimensional manifold with intrinsic dimension $d_0$, where $d_0\ll d$ ($d$ is the ambient dimension), convergence rate of our Alg 2 is minimax-optimal in-terms of $d_0$, without teh knowledge of $d_0$ is. In other words, our Alg 2 adapts to intrinsic dimension.
>
> Thank you for pointing out the typo in line 42, we will fix this. In line 537, we overload $\nu$ so that $\nu(r)$ denotes the smallest probability mass of any ball centered at $x$ with radius $r$. We will add a footnote to address this confusion. We do not see any incorrectness in the proof, in particular in lines 536-542, the required claim is given in Lemma D.1 which is a standard result from Chaudhuri and Dasgupta [2010]. However, we will spend some time to wordsmith this part of the proof so that it is more readable. Thank you for pointing out notational inconsistency in the proof of Lemma 3.5, we will fix this.

---

> > ### Comment · Reviewer_4EZA · 2024-08-10
> >
> > I thank the authors for their answers. They addressed my questions, so I increased my score accordingly.

---

> > > ### Author Response · Authors · 2024-08-12
> > >
> > > We sincerely thank the reviewer for increasing the score, we will integrate the clarification from the rebuttal into our revised version. Should there be any further questions or clarifications we can offer, please do not hesitate to let us know.

---

### Official Review · Reviewer_QQJv · 2024-07-09

**Soundness:** 3
**Presentation:** 3
**Contribution:** 3
**Rating:** 7
**Confidence:** 3

**Summary:**

The paper studies expand and sparsify the approach to non-parametric classification. The high-level idea is to first lift the example $x \in \mathcal{X} = \mathcal{S}^{d-1}$ to a $\{0,1\}^m$, a $m$-dimensional Boolean cube with exactly $k$ ones. The lifting is done by a linear mapping $x \mapsto \Theta x$ followed by top-$k$ selection using the magnitude of $\Theta x$, where the matrix $\Theta$ is sampled randomly. Let $\Theta x \mapsto z$ be the boolean feature obtained after the top-$k$ selection. The authors define a simple learning rule for the weight vector $w \in [0,1]^m$ and propose a linear classification rule $\mathbf{1}[w^{\top} x \geq k/2 ]$. Under the assumption that the score function $\eta: \mathcal{X} \to [0,1] $ of the Bayes optimal classifier is Holder-continuous, the proposed classifier is shown to be minimax optimal. The paper also proposes another classifier that can adapt if the example space $\mathcal{X}$ has a low-dimensional structure.

**Strengths:**

- The classifier is very natural and can be implemented.
- Experimental results show a strong performance almost on par with the nearest neighbor and random forest classifiers.
- The idea that one can lift the feature to a large enough boolean cube and run a sparse linear classifier is beautiful. Hopefully, there will be some followup works using this idea to solve other problems.

**Weaknesses:**

I did not find any apparent weakness in the paper. The paper delivers on its promise to provide a minimax optimal learner based on the expand and sparsify approach.

**Questions:**

Does Theorem 3.12 imply the consistency result in Theorem 3.3? If true, I suggest that the authors just state quantitative bound and infer consistency in a remark. The authors can use the saved space to give a more detailed proof sketch of Theorem 3.12. On the other hand, if the conditions for consistency in Theorem 3.3 and the quantitative rates in Theorem 3.12 are different, then some comments on that difference would be helpful.

---

> ### Author Rebuttal · Authors · 2024-08-05
>
> We thank the reviewer for providing encouraging feedback. We are in fact working on using this expand-and-sparsify representation idea to solve other ML problems.
>
> The question you have is an important one. The conditions for consistency in Theorem 3.3 and the quantitative rates in Theorem 3.12 are slightly different. The reason is, to establish consistency we simply used the classical Stone's theorem without caring much about the convergence rate (often for a new classifier, the first question one typically asks is whether the classifier is statistically consistent or not). To derive convergence rate, especially to establish minimax-optimality,  we did a bit more careful analysis. However, convergence rate does imply consistency and we will add a comment explaining this difference. If accepted, the camera ready version will provide one extra content page and we will add a proof sketch of Theorem 3.12 in the main paper as well.

---

> > ### Comment · Reviewer_QQJv · 2024-08-08
> >
> > I acknowledge the author's response and will retain my score. Thanks!

---

> > > ### Author Response · Authors · 2024-08-12
> > >
> > > We greatly appreciate your positive feedback. We will incorporate the clarifications into the revised version of our manuscript. Should there be any further questions or clarifications we can offer, please do not hesitate to let us know.

---

### Official Review · Reviewer_Q16i · 2024-07-11

**Soundness:** 4
**Presentation:** 3
**Contribution:** 2
**Rating:** 6
**Confidence:** 4

**Summary:**

The paper introduces two algorithms for non-parametric binary classification utilizing the expand-and-sparsify (EaS) representation. The first algorithm employs a winners-take-all approach for sparsification. It demonstrates consistency and achieves a minimax-optimal convergence rate that depends on the data dimension. The second algorithm is designed for scenarios where the data lies in a low-dimensional manifold. By utilizing the empirical k-thresholding operation, this algorithm attains the minimax optimal convergence rate dependent on the dimension of the low-dimensional manifold rather than the data dimension.

**Strengths:**

1. Relating non-parametric classification with expansion and EaS representation is both interesting and reasonable. The intuition behind it is straightforward: EaS offers a way to identify the 'neighbors' (active regions) of each data point, or in other words, points that are proximate to the data point in a certain representation space. By aggregating the conditional probability distribution of these 'neighbors,' we obtain a consistent classifier.

2. The proposed algorithm is simple and easy to implement. It leverages ideas from unsupervised learning representation and adapts them naturally to the supervised learning setting. It reminds me of several empirical machine learning algorithms, such as prototypical networks, which learn a representation for each class by computing the mean of the support examples' feature representation. During inference, the algorithms classify new examples based on the closest class prototype in the feature space. I believe this paper offers a way to conceptualize a class of empirically successful supervised learning algorithms, which involves learning representations and classifying data using examples that are close to it in the representation space.

3. The paper is well-organized and easy to read. The contributions, theorems, and underlying intuitions are presented clearly.

**Weaknesses:**

1. Most parts of the algorithms and some sections of their proofs come from [Dasgupta and Tosh, 2020] directly. The authors have pointed out that technically, they relax the Lipschitz continuous to Holder continuous and consider the effects of sample size. However, overall, the technical contribution is not significantly substantial in this regard. I would not view this point as a negative but rather as a neutral one.

2. I actually appreciate the way you have written the related work section, as it effectively illustrates the intuition behind EaS and how this idea has been developed. It also highlights the differences between this paper and previous works. However, there is a lack of related work on non-parametric classification, particularly focusing on their consistency and convergence results.

**Questions:**

1. As in Weakness 2, I suggest including more related work on non-parametric classification, particularly focusing on their consistency and convergence results.

2. In Algo 2, how do we choose t?

3. I appreciate your lower bound result in Theorem 3.15. However, we can modify Algo 1 by turning down the effects from $\theta_j$ that are far from $x$, specifically we can choose larger m and k, and then set eas[i] to be 0 when $\theta_i$ is not t-close to $x$. Does this modification make Algo 1 applicable to the manifold setting? If not, what is the intuition behind Algo 2 working in this scenario while the modified Algo 1 does not? In other words, how does 'empirical-k-thresholding' provide an advantage over 'k-WTA'?

4. The EaS method was initially proposed for unsupervised learning because it does not require labels. Given labeled data, we actually possess much more information. For instance, points with the same label should be closer in the representation space. However, your algorithm learns the representation and identifies the 'k-neighbors' primarily based on the data itself rather than their labels. From this perspective, I believe there could be modifications to the algorithm that better utilize the labels during the training phase. This could be a worthwhile area for further exploration.

## Minor:
1. In Algo 2, why we use first and second half of training data separately? In my understanding, the first half of the training data is used to determine the threshold, while the second half is used to determine $\eta_j$. So, we can probably use the whole batch of data twice to do the two things instead.
2. Page 3 line 137. $(x_1,y_1)$ should be $(x_i,y_i)$.
3. Page 4 line 153. It should be “An average of y values”.
4. Page 4 line 163. “Non-parametric”.
5. Page 8 line 323. “Alg. 2”: a missing “.”
6. Page 8 line 324. A missing $($.

**Limitations:**

The authors adequately addressed the limitations.

---

> ### Author Rebuttal · Authors · 2024-08-05
>
> We thank the reviewer for providing thoughtful and encouraging feedback. The reviewer's suggestion of including consistency and convergence rates of various non-parametric classification is an excellent one and if accepted we will definitely include a discussion on consistency and convergence rates of various non-parametric classifiers, such as, partitioning estimates (histograms), k-nearest neighbors, kernel methods and random forests.
>
> Regarding question 2, $t$ should be linear in $k$, for example $t=k/2$ or $t=k/4$, to have minimax-optimal convergence rate. However, it is possible to choose even $t=1$.
>
> Regarding question 3, your suggested modification of Alg1 is an interesting one. Note that in $k$-WTA sparsification, for any $x$, the closest $k$ $\theta_i$ are are activated (i.e., the corresponding bits in EaS representation are set to 1) since they have the largest $k$ dot products. Because of this property, your suggestion (choosing large $k$ first and then setting $eas[i]=0$ when $\theta_i$ is not $t$ close to $x$) boils down to choosing $k=t$. Intuitively, when data lie on or near a low dimensional manifold embedded  within a high dimensional unit sphere, and $\theta_i$ are chosen uniformly at random from the high dimensional unit sphere, only a tiny fraction of the $\theta_i$ will ever be used (activated for some $x$, i.e, corresponding bits in EaS representation will be 1), that is, they are one of the $k$ closest ones to at least one $x$ lying on the manifold. In other words, to have reasonable conditional probability estimates of the activated regions, which in turn may be used to provide a good estimate of $\eta(x)$, we may have to sample large number (exponential in $d$) of $\theta_i$s, in spite of the fact that most of them will possibly be unused (never activated at all)!
>
> In comparison, the empirical-k-thresholding is done in such a way that EaS representation of each $x$ is $k$-sparse in expectation but every single $\theta_i$ is used, that is,  activated for roughly $k/m$ fraction of the data points, (thus, reduces wastage as was the case for $k$-WTA sparsification), however, depending on the manifold structure, every single activated region $C_i$ may not be local anymore, and thus, $\eta(C_i)$ may not be reliable if we want to use it to estimate $\eta(x)$. For Alg 2, we first show in Lemma B.3 that when $\theta_i$ is  **good** in a sense that it is close to the manifold (please see Appendix B.3 for precise meaning) then $C_i$ is local in the sense that diameter of $C_i$ is small and depends only on $d_0$ instead of $d$ as was in the case of $k$-WTA sparsification. Then, we show in Lemma B.4 that when $\theta_i$ are chosen from appropriate Gaussian distribution, there are enough ``good" $\theta_i$ for each $x$ on the manifold with high probability. We set $t$ to be the number of  **good**  $\theta_i$ and for any $x$ on the manifold, the average of the $t$ corresponding conditional probability estimates $\eta(C_i)$ is a reasonable estimate of $\eta(x)$.
>
> Regarding question 4, your suggestion makes a lot of sense and we will definitely pursue this avenue in our future research.
>
> Finally, thank you so much for pointing out the typos. We will fix those. Regarding usage of two sets of data in Alg2, our motivation was to keep the estimated thresholds (computed from one set of data) independent of the other set used to estimate $\eta_j$. With a careful analysis, it may be possible to use only one set of data to compute both the thresholds and the $\eta_j$'s.

---

> > ### Comment · Reviewer_Q16i · 2024-08-07
> > **After Rebuttal**
> >
> > Thanks for the reply. The authors have addressed my questions. I will keep my score.

---

> > > ### Author Response · Authors · 2024-08-12
> > >
> > > We greatly appreciate your positive feedback. We will incorporate the clarifications into the revised version of our manuscript. Should there be any further questions or clarifications we can offer, please do not hesitate to let us know.

---

### Decision · Program_Chairs · 2024-09-25

**Decision:**

Accept (poster)

**Comment:**

Overall, reviews for this paper are positive -- reviewers feel that the results on expand-and-sparsify classifiers are interesting and novel and that the classifiers themselves are interesting and easy to implement. The paper is also very well-written. While the paper builds fairly directly on prior work of Dasgupta and Tosh [2020] it does make novel technical contributions, and is a good fit for NeurIPS.